# Convection-generated gravity waves in the tropical lower stratosphere from Aeolus wind profiling, GNSS-RO and ERA5 reanalysis

Mathieu Ratynski<sup>1,a,\*</sup>, Sergey Khaykin<sup>1</sup>, Alain Hauchecorne<sup>1</sup>, M. Joan Alexander<sup>2</sup>, Alexis Mariaccia<sup>3</sup>, Philippe Keckhut<sup>1</sup> and Antoine Mangin<sup>4</sup>

Correspondence to: Mathieu Ratynski (mathieu.ratynski@estaca.eu)

Abstract. The European Space Agency's Aeolus satellite, equipped with the Atmospheric LAser Doppler INstrument (ALADIN), provides near-global wind profiles from the surface to about 30 km altitude. These wind measurements enable the investigation of atmospheric dynamics, including gravity waves (GWs) in the upper troposphere and lower stratosphere (UTLS). This study analyzes ALADIN wind observations and ERA5 reanalysis, by deriving GWs kinetic energy (Ek) distributions, examining their temporal and spatial variability throughout the tropical UTLS. A prominent hotspot of enhanced GW activity is identified by Aeolus, migrating from the Indian Ocean in Boreal Summer to the Maritime Continent in Boreal Winter, closely matching outgoing longwave radiation minima and thus highlighting convective origins. Results show that ERA5 consistently underestimates Ek in convective regions, especially over the Indian Ocean, where conventional wind measurements are sparse. Additional comparisons with Global Navigation Satellite System Radio Occultation (GNSS-RO) measurements of GW potential energy (Ep) corroborate these findings and suggest significant underrepresentation of convection-driven wave activity in reanalyses. A multi-instrumental exploratory analysis also allows to verify the empirical grounding of the established Ek to Ep ratio. By providing direct wind measurements in otherwise data-sparse regions, Aeolus offers a valuable dataset for evaluating and potentially improving the representation of GWs in reanalyses, particularly in remote tropical areas. The combination of Aeolus and GNSS-RO data allows for an observationally-based examination of the partitioning between kinetic and potential energy, highlighting discrepancies with reanalysis products that could inform future model parameterization development.

<sup>&</sup>lt;sup>1</sup>Laboratoire Atmosphère Milieux et Observation Spatiales (LATMOS), UVSQ, CNRS, Univeristé Paris Saclay, Guyancourt, France

<sup>&</sup>lt;sup>2</sup>NorthWest Research Associate, Boulder, CO, US

<sup>&</sup>lt;sup>3</sup>Program in Atmospheric and Oceanic Sciences, Princeton University, Princeton, New Jersey, United States

<sup>&</sup>lt;sup>4</sup>ACRI-ST, 260 Route du Pin Montard, Sophia-Antipolis, Biot 06410, France

<sup>&</sup>lt;sup>a</sup>now at: Rosenstiel School of Marine and Atmospheric Science, University of Miami, Coral Gables, Florida, United States

# 1. Introduction

2 Atmospheric reanalyses like ERA5, a global atmospheric dataset produced by the European Centre for Medium-range Weather

3 Forecasts (ECMWF), are essential for climate assessments and atmospheric research (Hersbach et al., 2020). By integrating

4 observational data with state-of-the-art general circulation models and data assimilation methods, reanalyses provide

5 comprehensive atmospheric snapshots for a variety of meteorological research (Muñoz-Sabater et al., 2021).

However, one significant limitation of these datasets, including ERA5, is their reliance predominantly on temperature measurements for data assimilation, with wind measurements being notably sparse (Campos et al., 2022; Podglajen et al., 2014). Because of this, ERA5 tends to underestimate low-level wind speeds in certain regions, compared to radiosonde measurements (Munday et al., 2022). Having said that, only a relatively few radiosonde and cloud-tracked wind measurements directly constrain wind variability: Radiosonde measurements are notably sparse over oceans, as they are typically launched from land-based stations, leaving vast oceanic regions under-sampled (Baker et al., 2014; Ladstädter et al., 2011). While some ship-based radiosonde launches occur, they are infrequent and cover limited areas. Satellite cloud-tracking methods, such as Atmospheric Motion Vectors (AMVs), provide wind data by tracking cloud movements (Bedka et al., 2009). However, these methods have limitations: they cannot retrieve wind profiles in clear-sky conditions and often lack detailed vertical resolution. This results in significant observational gaps in wind measurements over oceans and clear-sky regions. This limitation is

and wind vertical profiles.

Gravity waves (GW) play a crucial role in the dynamics of the Earth's atmosphere. Generated by mechanisms such as flow over orography, convection, and flow deformation, these waves are instrumental in transporting momentum and energy, influencing atmospheric regions far from their origin points (Fritts and Alexander, 2003). While Rossby waves are well represented due to their quasi-geostrophic nature, divergent wave modes like gravity waves, Kelvin waves, Rossby-gravity waves, and inertia-gravity waves are not sufficiently characterized and must often be parametrized internally by the models (Plougonven and Zhang, 2014). The underrepresentation of gravity waves with long horizontal and short vertical scales in ERA5 has been highlighted previously (Bramberger et al., 2022).

particularly critical when considering atmospheric waves, such as gravity waves, which manifest themselves in temperature

For the study period from June 2019 to August 2022, ERA5 utilizes the non-orographic gravity wave drag (GWD) scheme described by Orr et al., (2010), which is based on a spectral approach (Scinocca, 2003). This scheme does not explicitly resolve convectively generated waves based on model-diagnosed convection; instead, it launches a globally uniform and constant spectrum of waves from the troposphere. The momentum deposition occurs as these waves propagate vertically and interact with the resolved flow via critical-level filtering and nonlinear dissipation. While this parameterization improves the middle

atmosphere climate compared to simpler schemes, evaluations have shown it has limitations in fully capturing the required wave forcing, particularly for the Quasi-Biennial Oscillation (QBO) in the tropics (Pahlavan et al., 2021).

Furthermore, even with improvements in reanalysis products, challenges in accurately representing tropical winds persist. Studies of previous-generation reanalyses identified significant errors in tropical regions (Podglajen et al., 2014), and recent work shows that even ERA5's accuracy is highly site-dependent, with notable errors in locations influenced by warm currents (Campos et al., 2022). This is compounded by difficulties in data assimilation systems, such as 4-D var and perfect model scenarios, which struggle to extract circulation information from high-resolution temperature data (Žagar et al., 2004). Despite advancements in the quality of tropical forecasts and analyses, the evidence suggests that radio occultation (RO) data could potentially enable effective long-term monitoring of wind fields globally (Danzer et al., 2023). However, the overall lack of direct wind observations continues to pose significant challenges (Baker et al., 2014).

Historically, most GW studies have relied on ground-based or single-use instruments like radiosondes (Zhang and Yi, 2005), rockets (Wüst and Bittner, 2008), or global coverage measurements from the Global Navigation Satellite System Radio Occultation (GNSS-RO). While GNSS-RO provides high-resolution temperature profiling, effectively characterizing GW potential energy (Ep) (Fröhlich et al., 2007; Khaykin et al., 2015; Schmidt et al., 2016), it does not capture kinetic energy (Ek), which requires precise wind profiling.

In an effort to bridge many gaps within the observational world, the 2018 launch of the European Space Agency's Aeolus satellite changed our ability to capture atmospheric dynamics, particularly in the upper troposphere and lower stratosphere (UTLS). The UTLS is a region marked by a dramatic increase in static stability at the tropopause, where gravity waves are refracted to shorter vertical wavelengths (Dhaka et al., 2006; Geldenhuys et al., 2023). These waves with short vertical wavelengths (typically 2-10 km) are primarily lower-frequency gravity waves, as dictated by the dispersion relation, and exhibit relatively large amplitude wind variability. The Aeolus satellite, equipped with its Atmospheric LAser Doppler INstrument (ALADIN), is able to measure global wind profiles up to an altitude of 30 km, providing insights into the behavior of gravity waves with vertical wavelengths down to ~1.5-2 km in these critical atmospheric layers (Banyard et al., 2021; Rennie et al., 2021; Ratynski et al., 2023).

In this context, this study aims at utilizing Aeolus's global wind profiling capabilities to derive a tropics-wide distribution and variability of the kinetic energy of gravity waves, addressing a gap not typically captured in ERA5 reanalysis. By comparing direct measurements with ERA5 data, we reveal certain limitations in the reanalysis's ability to represent tropical gravity wave dynamics. We will look at the most recent reprocessed Aeolus baseline 2B16, providing data from June 2019 to August 2022. Additionally, our study aims at exploring a broader set of analyses, aiming to contextualize the Aeolus wind observations within a multi-instrument framework. By comparing Aeolus-derived kinetic energy of GWs with the potential energy estimates

from GNSS-RO, we assess the consistency of independent data sources and examine the ratio of kinetic to potential energy in real-world atmospheric conditions. With this study, we provide the first observationally-based, tropics-wide estimate of gravity wave kinetic energy from June 2019 to August 2022, directly linking its variability to deep convective sources.

The paper will be organized as follows: In Sect. 2, we will discuss the data as well as the methods. It includes a description of the Aeolus, ERA5 and the GNSS-RO datasets, but also explains the horizontal detrending method with its potential and limitations. In Sect. 3, we will analyze the wave activity in terms of kinetic energy using Aeolus Rayleigh wind profiling and directly comparing it with ERA5. Additionally, in Sect. 4, we broaden our analyses to contextualize Aeolus observations against GNSS-RO data and criticize the ratio between both elements. Finally, the results are discussed in Sect. 5, followed by the conclusions in Sect. 6

# 2. Data and Methods

# 2.1 Instruments and Datasets

The Aeolus satellite, with its ALADIN Doppler wind lidar, orbited Earth at a 97-degree inclination and 320 km altitude. Its data consists of 24 vertical range bins that divide the atmosphere, allowing wind profiling between 0 and 30 km. Laser pulses and two receivers—Rayleigh and Mie channels—detect the atmosphere's Doppler shifts through molecular and particle backscatter, respectively. The data, organized into atmospheric scenes, cloudy or clear, has an 87 km along-track integration and a vertical resolution varying between 0.25 to 2 km. Within the tropical UTLS region of this study, the vertical bin size is typically between 0.5 and 1.5 km. The distribution of these range bins is determined by a dedicated range bin setting (RBS), which varies geographically to meet different observational goals, with distinct configurations routinely used for the tropics, extratropics, and polar regions. This study uses the Level 2B Rayleigh clear product from June 2019 to August 2022, with the latest Baseline 2B16 at the time of submission, offering the horizontal line of sight (HLOS) wind components. The HLOS wind speed is derived using Aeolus NWP Impact Experiments guidance, with the vertical wind speed assumed to be negligible. A complete description of the instrument, its measurement principles, range bin settings, and data products can be found in Rennie and Isaksen. (2024). The angle  $\theta$  denotes the azimuth of the target-to-satellite pointing vector, being around 100.5° over the tropics. When injecting the azimuth value into Eq. 1, it becomes apparent that the HLOS wind over the tropics is quasi-zonal.

94 
$$v_{HLOS} = -u \sin(\theta) - v \cos(\theta)$$
 (1)

The ERA5 reanalysis dataset, a ECMWF product, offers comprehensive atmospheric, land-surface, and ocean-wave parameters at hourly resolution and global coverage (Hersbach et al., 2020). Its exceptional horizontal resolution of

approximately 33 km at the equator (corresponding to 0.3° latitude/longitude), the best among widely used reanalysis products, enables it to resolve gravity waves with horizontal wavelengths as small as ~100 km (Wright and Hindley, 2018, their table 1). The data products used in this study were retrieved from the ECMWF archive on a regular 0.25° x 0.25° latitude-longitude grid. Additionally, its higher vertical resolution in the troposphere, with 137 vertical levels reaching up to 0.01 hPa, makes it particularly adept at capturing gravity waves with vertical wavelengths down to ~1–2 km. ERA5 also incorporates advanced modelling features such as sponge layers and hyperdiffusion to attenuate artificial wave reflections and stabilize the model numerically, allowing for efficient modelling of large-scale phenomena, notably simulating gravity waves with wavelengths greater than 400 km (Stephan and Mariaccia., 2021). It is therefore the best candidate to use as a benchmark for Aeolus' performances. For representing sub-grid scale gravity waves, the ERA5 configuration used in this study employs a non-orographic GWD parameterization that is not directly forced by model-diagnosed convection (Orr et al., 2010). Instead, the scheme launches a globally uniform spectrum of waves from the troposphere. For this study, wind components are retrieved on the native 137 model levels. To prepare the data for analysis, the geopotential height of each model level is first converted to geometric altitude. The vertical profiles are then linearly interpolated from this native geometric altitude grid onto the standard 100 m high-resolution grid used for all datasets in this study.

The GNSS-RO method offers many advantages for studying atmospheric dynamics, particularly GW activity and parameters. The first RO-derived GW estimates date back from the early 2000s and several missions have since provided data for further studies, focusing on potential energy as a proxy for retrieving GW activity (Tsuda et al., 2000; Fröhlich et al., 2007; Wang and Alexander, 2010; Luna et al., 2013; Schmidt et al., 2016). The Radio Occultation Meteorology Satellite Application Facility (ROMSAF) provides global GNSS-RO datasets. For the study period of June 2019 to August 2022 these datasets are dominated by the Metop constellation: Metop-B and Metop-C throughout, with Metop-A contributing until its retirement in November 2021 (von Engeln et al., 2011). These datasets are derived from the bending angles of GNSS signals as they pass through the Earth's atmosphere and are observed by low Earth-orbiting satellites. It provides global coverage with a high vertical resolution, sub-Kelvin accuracy, full diurnal coverage, and all-weather capability. The vertical resolution of GNSS-RO temperature profiles is fundamentally limited by diffraction and varies with altitude, typically ranging from ~0.5 km in the lower troposphere to ~1.4 km in the middle atmosphere (Kursinski et al., 1997). While sharp vertical gradients in refractivity (e.g., due to temperature inversions or strong humidity gradients) can be detected, the effective resolution for resolving distinct atmospheric layers is constrained by these diffraction limits. The along-track horizontal resolution is typically around 200-300 km. Marquart and Healy (2005) showed that small-scale fluctuations in dry temperature RO profiles could be attributed to GWs with vertical wavelengths equal to or greater than 2 kilometers. Alexander et al. (2008b) suggested analyzing data below 30 kilometers in altitude to maintain the signal-to-noise ratio for temperature fluctuations above the detection threshold, which also happens to be Aeolus' maximal capability. Most GW parameters can be derived from single RO temperature profiles. However, estimating momentum flux requires knowledge of the horizontal wave number or wavelength, which cannot be deduced from a single temperature profile. To determine the horizontal structure of GWs, it is necessary to analyze clusters of three or more profiles adjacent in space and time (Schmidt et al., 2016).

131132

130

- This study specifically utilizes Aeolus Level 2B Rayleigh clear HLOS winds, ERA5 wind components, and GNSS-RO temperature profiles, all brought to a standard interpolated grid to facilitate the accurate comparison and integration of data from the different sources. The chosen grid has a vertical resolution of 100 meters and spans a range from 0 to 30 km altitude.
- This approach preserves the maximum vertical detail from each dataset before analysis.

The choice to compare Aeolus measurements with the ERA5 reanalysis, which does not assimilate Aeolus winds serves as a comparison with an independent dataset. This approach thereby highlights regions where its direct wind measurements might fill observational gaps present in the conventional observing system assimilated by ERA5.

## 2.2 Methods and Limitations

- The following section discusses the retrieval of GW kinetic energy, Ek. A primary challenge in this retrieval, particularly in the tropical UTLS, is the robust separation of GWs from other dominant, synoptic-to-planetary scale equatorial waves, such as Kelvin waves. Observational studies using GNSS-RO data have consistently shown that Kelvin waves, with typical vertical wavelengths in the range of ~4-8 km (Randel et al., 2021; Randel and Wu, 2005), are a prominent feature of the tropical temperature and wind fields. This presents a potential for spectral overlap with the longer vertical wavelength portion of the GW spectrum that this study aims to capture.
- Several methods exist for background state determination and large-scale process separation. These broadly fall into two categories: Vertical Detrending (VD), often applied to single profiles from instruments like lidars and radiosondes (Gubenko et al., 2012; Khaykin et al., 2015), and Horizontal Detrending (HD). HD requires spatially resolved datasets like satellite observations or model reanalyses and typically involves spatio-temporal averaging to define the background (Alexander et al., 2008b; Khaykin et al., 2015). A comparative study by John and Kumar., (2013) highlighted significant discrepancies in derived Ep magnitudes depending on the chosen method.
- The choice of detrending method is particularly critical in the tropics due to the presence of waves like Kelvin waves. VD methods, if not carefully designed, may inadvertently remove GWs with long vertical wavelengths or, conversely, retain short vertical wavelength components of planetary-scale waves (e.g., Kelvin waves observed with vertical wavelengths as short as 3 km, as noted by Alexander and Ortland., (2010) and Cao et al., (2022)). Consequently, Schmidt et al. (2016) strongly recommend using HD for satellite data, as VD may overestimate GW activity by including remnant signals from synoptic and

planetary waves that possess significant vertical structure in the tropics. Given these considerations, our study employs an HD approach, calculating the background profile within a fixed spatio-temporal grid (20° longitude × 5° latitude over 7 days), which we deem best suited for retrieving GW energy information from the Aeolus, GNSS-RO, and ERA5 datasets.

The separation of the wind or temperature profile into a background state and perturbations using HD is intended to isolate fluctuations characteristic of gravity waves by filtering out larger-scale and slower-evolving processes like the mean components of Rossby and Kelvin waves. This selection relies on the distinct scale and structural characteristics of GW perturbations. However, the work by Randel et al., (2021) using dense COSMIC-2 RO data reveals further complexities. They found that "residual" small-scale temperature variances (analogous to our perturbation fields) exhibit coherent maxima in the longitudinal and vertical shear zones of large-scale Kelvin waves. This suggests that the local atmospheric environment shaped by Kelvin waves, particularly variations in static stability (N²), can modulate the amplitude of smaller-scale variability, potentially including GWs. Furthermore, data assimilation studies have demonstrated that the inclusion of Aeolus wind data directly impacts the representation of vertically propagating Kelvin waves in numerical weather prediction models. This impact is explicitly linked to the background wind, with the largest analysis changes occurring in regions of strong vertical wind shear (Žagar et al., 2021, 2025). This highlights the importance of direct wind observations in these critical regions. Indeed, direct analysis of Aeolus observations (without assimilation) confirms that Kelvin waves are well-resolved, showing good agreement in wave variances when compared to reanalyses (Ern et al., 2023). This implies that the characteristics of Kelvin waves seen by Aeolus are robust and may differ from those in reanalyses not assimilating Aeolus data.

To further refine the isolation of GWs and address the potential aliasing from such equatorial waves, our HD approach is combined with a vertical band-pass filter applied to the perturbation profiles. This filter targets vertical wavelengths between 1.5 km and 9 km. The lower limit is chosen based on the effective vertical resolution of the instruments (particularly Aeolus), while the upper limit of 9 km is selected to be slightly above the typical dominant vertical wavelengths reported for Kelvin waves in the UTLS, thereby further reducing their potential contribution. This combined HD and vertical filtering methodology has been widely used for retrieving GW Ep from temperature data (Alexander et al., 2008b; Schmidt et al., 2008; Šácha et al., 2014; Khaykin et al., 2015), and the availability of Aeolus wind profiles allows us to apply a consistent approach for GW Ek.

Nevertheless, it is acknowledged that a perfect separation is difficult. The sharpest vertical gradients associated with Kelvin waves, or localized enhancements of GW activity within Kelvin wave-modified environments as suggested by Randel et al., (2021) might still contribute to the derived GW energy. The interpretation of our GW Ek and Ep must therefore consider this context, particularly when analyzing variability in regions known for strong equatorial wave activity.

Figure 1. Derivation of GW energy profiles from wind measurements (a) Observed wind profile and the corresponding background state profile. (b) Wind perturbation profile alongside its filtered counterpart. (c) Resulting in Ek, smoothed and then averaged within the given altitude range.

Based on the linear theory of GW, the measured wind profile U(z) shown in Fig.1a is divided into a background wind  $\overline{U}(z)$  also present in Fig.1a and a perturbation U'(z) depicted in Fig.1b. The background is obtained by averaging all individual wind profiles for kinetic energy retrieval, within a spatiotemporal grid box of 20° longitude  $\times$  5° latitude over 7 days. While this horizontal detrending method was originally demonstrated using temperature profiles in Alexander et al., (2008b), its application to wind profiles is theoretically sound. Linear gravity wave theory dictates that wind and temperature perturbations are coupled manifestations of the same wave phenomena, and thus the principle of separating smaller-scale waves from the large-scale background flow via spatiotemporal averaging is equally valid for both fields. Following the arguments presented in Alexander et al., (2008b), this choice is justified by the need to ensure a sufficient number of profiles per grid cell, which minimizes random noise while preserving meaningful variability in the data. Shorter temporal windows would lead to insufficient sampling, while longer windows would smooth out critical small-scale wave features. The grid size is also designed to preserve the spatiotemporal variability of mesoscale gravity waves and equatorially trapped structures, in an attempt to separate the background and perturbation components without introducing significant biases.

Finally, this configuration mitigates errors in the definition of the  $\overline{U}(z)$  profile, ensuring reliable kinetic energy calculations and robust separation of gravity wave perturbations. We performed sensitivity tests with varying grid sizes and temporal windows to confirm that this configuration provides the best possible background state when prioritizing Aeolus retrieval (see Fig. A1 in Appendix A). The average number of profiles used for the background state determination is 55 for Aeolus, 20 for GNSS-RO and 1400 for ERA5.

The next step involves subtracting the background profile from its corresponding individual profile, eliminating most large-scale waves (Planetary Waves, Kelvin Waves, Rossby Waves). This yields the perturbation profile U'(z), which is then subjected to Welch-windowing, which is done in order to mitigate spectral leakage (Alexander et al., 2008a; 2008b; Khaykin et al., 2015). A prior study also applied a similar windowing function (half cosine), aiming to counteract the "effects of the edge of the height range" (Hei et al., 2008). After said windowing, a band-pass filter designed to retain vertical wavelengths between 1.5 km and 9 km. is applied to the perturbation profile, as seen in Fig.1b and 1c. The upper limit of 9 km isolates GWs from larger-scale planetary waves, consistent with our background removal strategy. The lower limit of 1.5 km is chosen to reflect the effective vertical resolution of the Aeolus instrument (Ratynski et al., 2023) and ensures that our comparison is restricted to wave scales reliably resolved by all datasets (Banyard et al., 2021). This procedure provides a methodologically consistent basis for comparing GW energy across the different instruments.

The GW Ek can be derived from the variance of wind components as follows:

25 
$$E_k = \frac{1}{2} \left( \overline{u'^2} + \overline{v'^2} + \overline{w'^2} \right),$$
 (2)

where u, v, and w represent the zonal, meridional, and vertical wind components, respectively. Considering that Aeolus's viewing geometry in the tropics makes its HLOS wind primarily sensitive to the zonal component (as shown in Eq. 1), we will note all mentions of retrieved speed as *u* for clarity. In our case, since the vertical wind speed is neglected and the satellite is not able to distinguish between zonal and meridional wind, it is necessary to provide a new formalism for the retrieved metric:

$$231 E_{k HLOS} = \frac{1}{2} \left( \overline{u'^2}_{HLOS} \right) (3)$$

The resulting profile, which is essentially the perturbation squared, is cut to keep the data between one kilometer below the tropopause and 22 km. The altitude range is chosen considering Aeolus' limitations, such as increasing error at higher altitudes due to lack of backscatter signal (Ratynski et al., 2023, their Fig.3). For consistency, the tropopause height is derived directly from the ERA5 dataset for all analyses. The tropopause is calculated for each profile based on the WMO thermal definition, where the tropopause is the lowest level at which the lapse rate decreases to 2 K/km or less, provided the average lapse rate

within the 2 km layer above remains below this threshold. The profile is then averaged over the selected range, representing the Ek, as seen in Fig.1c.

We acknowledge that including the layer just below the tropopause presents a potential challenge, as strong, non-wave divergent outflow from deep convection could be partially aliased into our derived kinetic energy (Stephan et al., 2021). To rigorously test the robustness of our results against this potential contamination, we have performed a comprehensive sensitivity analysis by recalculating the kinetic energy fields using two more conservative averaging layers, starting from 1 km and 2 km above the tropopause, respectively (see Fig.B1 and B2 in Appendix B). By shifting the analysis layer upward to such levels, we confirm that the geographical patterns of the energy hotspots are remarkably stable (spatial correlation r > 0.83), and that the vast majority of the peak energy (~88-91%) persists well into the stratosphere. If the signal were dominated by shallow tropospheric outflow, the energy peaks would have collapsed when the analysis layer was moved above the tropopause. The fact that a strong, structured signal remains confirms that our method is observing vertically propagating gravity waves that have penetrated the lower stratosphere.

Although the above steps focus on retrieving GW Ek from Aeolus wind measurements, the same procedure can be applied to temperature-based observations such as GNSS-RO for Ep. The main difference lies in substituting temperature T(z) for wind U(z) throughout the background-perturbation decomposition, which means using T'(z) rather than U'(z). The Welch window was applied to all perturbation profiles (wind and temperature) before filtering to mitigate spectral leakage. The same band-pass filtering strategy and vertical averaging then provide the Ep profile from the temperature perturbations. In this case, the GW Ep is calculated using this formula:

$$E_p = \frac{1}{2} \left(\frac{g}{N}\right)^2 \left(\frac{T'}{T}\right)^2 \tag{4}$$

The  $E_{k\,HLOS}$  metric derived from Aeolus (Eq. 3) represents the kinetic energy projected onto the instrument's line of sight. Since our study focuses on the tropical UTLS region, the meridional wind component will have a minor contribution compared to the zonal component. Therefore, the  $E_{k\,HLOS}$  energy represents primarily the zonal activity, meaning that we are missing a non-negligible proportion of wave activity. To evaluate the contribution of v' to the total kinetic energy we use ERA5 data and compute the ratio between total Ek (derived from u' and v') and  $E_{k\,HLOS}$  (as it is observed by ALADIN).

This ratio (Fig. C1 in Appendix C1) exhibits significant geographic variability, which can be linked to physical mechanisms that create wave anisotropy. For instance, over regions like the Indian Ocean, the ratio is relatively low (~1.5), suggesting a predominantly zonal orientation of wave energy. This is physically plausible, as persistent surface winds like the trade winds can influence the tropopause-level wave field through two main processes. Firstly, flow over orography can preferentially generate zonally-oriented waves (Kruse et al., 2023). Secondly, the background wind profile itself acts as a directional filter,

selectively allowing waves propagating in certain directions to reach the UTLS while attenuating others through critical-level interactions (Plougonven et al., 2017; Achatz et al., 2024).

When averaged over the mission period and focused on the equatorial band ( $10^{\circ}\text{S}-10^{\circ}\text{N}$ ), the ratio settles at approximately 1.6. This implies that  $Ek_{HLOS}$  accounts for around 62.5% of total Ek, the remainder being undetectable due to HLOS projection. The meridional component, less significant in this specific geographical area for Aeolus, contributes the remaining 37.5% of Ek not considered by  $Ek_{HLOS}$ . Although not dominant,  $Ek_{HLOS}$  represent a substantial contribution to Ek. This scaling factor is used when discussing the implications of our Aeolus findings for the total GW kinetic energy budget. The details of the spatio-temporal variability of this ratio are provided in the Appendix C.

The methods employed constrain the analysis to a specific range of horizontal and vertical wavelengths. Aeolus' RBS determines the spacing between sampling points, impacting the vertical and horizontal resolution and maximal detectable wavelength. The vertical wavelength analysis is constrained by the 9 km upper band-pass, representing roughly half the average profile length in the tropics, after limiting the profile to the optimal range and especially considering the dynamic lower bound. Profiles generally extend to heights between 23km and 26km. In the horizontal dimension, since a 20° x 5° degrees grid is used for the background removal and the wind is supposed quasi-zonal, the zonal wavelengths, therefore, reside below 2220 km.

Additionally, Aeolus can be prone to errors alternating the quality of wind profiles. Amongst the most notable ones are dark currents in the charge-coupled devices ("hot pixels"), potentially leading to errors of up to several meters per second (Weiler et al., 2021). Another identified issue is the oscillating perturbations, parasitic deformations of the signal, yet to be attributed to a cause, which can be mistaken for GW-induced signals (Ratynski et al., 2023). While corrections were implemented for the first issue (Weiler et al., 2021), the overall signal random error varies with time, with a general tendency to increase due to instrument degradation (Lux et al., 2022). Aeolus' HLOS wind variance is inherently linked to the measurement noise (i.e., random error). In other words, the observed wind variance is a sum of the variance due to waves (detected using the given data and method) and the variance due to ALADIN noise, i.e., its random error squared.

$$296 \quad \overline{u'^2}_{HLOS} = \overline{u'^2}_{GW} + \overline{u'^2}_{LN} \tag{5}$$

with  $\overline{u'^2}_{GW}$  representing the variance contribution from gravity waves and  $\overline{u'^2}_{I.N}$  the contribution from instrument noise. Since kinetic energy is proportional to variance, this relationship holds for kinetic energy as well. The observed Aeolus kinetic energy  $\overline{Ek_{Aeolus\,HLOS}}$  is therefore the sum of the true geophysical signal and a noise component which increases over the mission lifetime. To isolate the true gravity wave energy, this time-varying noise component must be estimated and removed. This correction is performed at the kinetic energy level. While radiosondes provide a valuable independent reference, their sparse

coverage in the tropics makes them unsuitable for creating a globally consistent correction field. We therefore use the ERA5 reanalysis as a temporally stable global reference to estimate the Aeolus instrument noise. The core principle is to produce a corrected dataset, denoted as  $\overline{Ek_{Aeolus\,HLOS*}}$  by subtracting our best estimate of the noise energy  $\overline{Ek_{I.N}}$  from the observed energy:

$$\overline{Ek_{Aeolus\,HLOS*}} = \overline{Ek_{Aeolus\,HLOS}} - \overline{Ek_{I.N}}$$
 (6)

hotspots uniquely captured by Aeolus, or over-correcting areas of low variance.

- The estimation of the noise term,  $\overline{Ek_{I.N}}$ , is not a simple subtraction. It is derived using a spatio-temporally adaptive algorithm that blends an additive offset (representing baseline instrument noise, dominant in quiescent atmospheric regions) with a multiplicative scaling factor (more influential in active convective regions where noise effects might scale with the signal). This adaptive approach ensures that instrumental artefacts are removed without suppressing the high-energy gravity wave
- An additional refinement step is required for seasonally averaged geographical maps. To produce these maps, individual profile energy values were first binned into a 5° longitude by 2° latitude grid. A second stage adapts the noise correction derived from the tropical 10°S–10°N band for application to the broader latitudinal extent of the maps (e.g., 30°S–30°N), accounting for latitudinal variations in energy and ensuring physically consistent, non-negative results. To reduce noise and highlight large-scale patterns, a 3-point median filter followed by a 3-point moving average filter was applied sequentially in both the zonal and meridional directions.
- The full mathematical derivation of the adaptive estimation of  $\overline{Ek_{I.N}}$ , the details of the map-specific refinement, and a series of diagnostic plots, including comparisons of Aeolus data before and after correction to validate the assumptions made, are provided in the Appendix D.

#### 3. Results 322

# 3.1 Seasonal variation of GW Ek

Figure 2. Comparison between Ek<sub>Aeolus HLOS</sub>\* (left column) and Ek<sub>ERA5 HLOS</sub> (right column). Each row corresponds to a season, from June-July-August 2019 to March-April-May 2021. The UTLS altitudes are defined between one kilometer below the tropopause and 22 km. The tropopause is determined from the ERA5 reanalysis. The maps are smoothed using a combination of median and moving average filters as described in the Methods section.

Figure 2 displays the Ek<sub>HLOS</sub> distribution from JJA 2019 to MAM 2021, derived from the corrected Aeolus observations and the ERA5 reanalysis. This comparison reveals both key similarities in two large-scale patterns: first, the confinement of most GW kinetic energy to the equatorial belt (approximately 15°S–15°N), and second, a distinct seasonal migration of this energy. However, there are also significant differences in the representation of regional wave activity.

Both datasets consistently show that the majority of GW kinetic energy is confined to the equatorial belt (approximately 15°S–15°N). This observation aligns with the expectation that deep tropical convection, concentrated within the Intertropical Convergence Zone (ITCZ), is a primary source of the observed waves. The reader should note that some variance from other equatorial waves, centered at the equator by definition, will also be inevitably present to a small degree in the perturbation fields. A clear seasonal cycle is evident in both Aeolus and ERA5. During Boreal summer (JJA), enhanced Ek is prominent over Central Africa and the Indian Ocean. This corresponds to the active phases of the African and Indian monsoon systems, which provide a persistent, large-scale environment favorable for the development of organized, deep convective systems known to be efficient gravity wave generators (Forbes et al., 2022). During Boreal winter (DJF), the focus of activity shifts eastward towards the Maritime Continent and the Western Pacific, coinciding with that region's primary convective season. These general patterns are also consistent with previous climatologies of GW potential energy derived from temperature measurements Alexander et al. (2008b, their Fig.3 and Fig.4). The GNSS-RO derived Ep values, which range from 0 to 6.6 J/kg at 15 km and 0 to 4.4 J/kg at 22 km (Alexander et al., 2008c), after applying the usual Ek/Ep ratio of 1.6, are generally aligned with our observations.

It is also necessary to clarify the interpretation of the wave activity observed at the subtropical edges of our analysis domain (near 30°N/S). While our study focuses on convectively generated waves originating from the deep tropics, the kinetic energy measured in the subtropics is likely dominated by different, local sources. The strong subtropical jets and associated frontal systems are potent generators of inertia-gravity waves through mechanisms of geostrophic adjustment and shear instability (Kruse et al., 2023; Plougonven and Zhang, 2014). A recent case study has confirmed that such jet-merging events can produce significant, large-scale GW fields (Woiwode et al., 2023). These jet- and front-generated waves typically have sub-weekly periods and significant wind perturbations, meaning they fall within the detection window of our filtering methodology (Achatz et al., 2024). Therefore, the enhanced energy often visible near 30°N and 30°S in our seasonal maps should be interpreted as stemming primarily from these midlatitude dynamical processes, rather than from the poleward propagation of the equatorial convective waves. These jet- and front-generated waves are dynamically distinct from the deep tropical convection associated with the major seasonal monsoon systems. While the subtropical jets produce notable GW activity, our results indicate that the most intense and geographically extensive hotspots are found within the equatorial belt and are closely tied to these monsoon systems (Kang et al., 2017; Wright and Gille, 2011).

Despite these general agreements, a critical difference emerges in the structure and intensity of the energy hotspots. ERA5 tends to represent GW activity as a relatively smooth, zonally elongated band, with modest seasonal modulation and appears to significantly miss wave activity both in the active monsoon regions and in more structured events further from the equator. In stark contrast, Aeolus reveals a picture of much more localized and intense Ek hotspots. For example, during JJA 2020 and SON 2020, Aeolus observes a well-defined hotspot over the Indian Ocean with Ek values exceeding 10-12 J/kg, whereas ERA5 shows only a diffuse enhancement in the same region with values rarely exceeding 5-7 J/kg. Similarly, the DJF 2020/21 hotspot over the Maritime Continent is markedly stronger and more geographically confined in the Aeolus data.

This discrepancy suggests that while ERA5 captures the broad climatic envelope of convective GW activity, it significantly underestimates the peak energy of waves generated by localized, intense convective systems. This is particularly evident in regions where conventional wind observations are sparse, such as the Indian Ocean. The direct wind profiles from Aeolus appear to capture magnitudes and structures of this convection-driven wave activity that are not present in the reanalysis.

The period from mid-2020 onward, which coincided with the development of La Niña conditions, exhibits the most pronounced differences between the two datasets. While La Niña is known to enhance convection over the Maritime Continent, the consistently higher energy levels observed by Aeolus across all regions during this period also correlate with a documented increase in the satellite's instrumental random error (Ratynski et al., 2023, their Fig.6). Our adaptive noise correction (see Appendix D) is designed to account for this degradation. However, it is challenging to perfectly disentangle the increased geophysical signal (e.g., from La Niña) from the effects of increased instrument noise. Nevertheless, the geographical consistency of the hotspots observed by Aeolus, which align with known convective centers, provides confidence that the primary patterns represent true atmospheric phenomena that are underrepresented in the reanalysis. The direct link between these kinetic energy hotspots and deep convection will be examined in detail in the following section through a comparison with Outgoing Longwave Radiation data.

Finally, regarding the strong latitudinal confinement of the signal, while this is primarily a physical feature, our noise-correction methodology may also contribute to it. As detailed in the Appendix D (Part 2), the correction is weighted by the latitudinal structure of the raw signal. This approach, designed to avoid over-correction in low-signal subtropical regions, naturally sharpens the latitudinal gradient at the edges of the tropical belt.

# 3.2 Zonal variation of GW activity from observations and ERA5

394

396

401

404

#### a) Aeolus HLOS GW Ek - UT/LS | 10S - 10N b) ERA5 HLOS GW Ek - UT/LS 10S - 10N c) Aeolus - ERA5 GW Ek - UT/LS | 10S - 10N 01-Sep-2022 01-Sep-2022 01-Sep-2022 13.5 13.5 01-Jun-2022 01-Jun-2022 01-Jun-2022 01-Mar-2022 01-Mar-2022 01-Mar-2022 12 12 01-Dec-2021 01-Dec-2021 01-Dec-202 10.5 10.5 01-Sep-2021 01-Sep-2021 01-Sep-2021 01-Jun-2021 01-Jun-2021 01-Jun-2021 01-Mar-2021 01-Mar-2021 01-Mar-2021 7.5 01-Dec-2020 01-Dec-2020 01-Dec-2020 01-Sep-2020 01-Sep-2020 01-Sep-2020 01-Jun-2020 01-Jun-2020 45 01-Jun-2020 4.5 01-Mar-2020 01-Mar-2020 01-Mar-2020 01-Dec-2019 1.5 1.5 01-Sep-2019 01-Sep-2019 01-Sep-2019 01-Jun-2019 01-Jun-2019 01-Jun-2019 120 180 240 120 180 240 120 180 240 300 360 60 300 360 30 Longitude (°E) Longitude (°E) Longitude (°E)

Figure 3. (a,b,c) Hovmoller diagram of  $Ek_{Aeolus\ HLOS^*}$ ,  $Ek_{ERA5\ HLOS}$  and their difference. The contour plot represents the Outgoing Longwave Radiation (OLR) for 210 and 220 W/m² (black and white, respectively). Each bin corresponds to an average of over 3 weeks and 10 degrees. The white bins represent the lack of satellite information in (a). The OLR measurements were obtained from the Australian Bureau of Meteorology. The UTLS altitudes are defined between one kilometer below the tropopause and 22 km. The tropopause is determined from the ERA5 reanalysis. Black stippling indicates regions where the difference between quantities is statistically significant (two-sample t-test, p < 0.05).

To assess the evolution and transition between the different seasons with greater precision, the Hovmoller diagrams in Fig.3 only show the HLOS-projected GW kinetic energy from Aeolus and ERA5, along with their difference within the deep tropics between 10° N and 10° S, as Fig.2 proves this region contains most of the activity. To identify regions of deep convection, these diagrams are overlaid with contours of low Outgoing Longwave Radiation (OLR), a reliable proxy for deep convection as it indicates cold, high-altitude cloud tops and thus the depth of convective systems (Zhang et al., 2017).

Fig.3a shows Ek<sub>Aeolus HLOS\*</sub>, where prominent hotspots of high Ek (often attaining 15 J/kg) are visible, with a broad region of enhanced activity migrating eastward from the African continent (~0-60°E) towards the Indian Ocean and Maritime Continent (~60-150°E) between June and March. This shift is recurring over multiple years and shows a relative consistency between each year in terms of longitudinal and temporal range. This migration of high Ek is systematically co-located with the seasonal cycle of convection, with the hotspots consistently falling within the low OLR contours (below 220 W/m²).

The presence of hotspots, represented by distinct shapes in the Ek patterns, is expected in regions with prevalent convective 407 activity. These can be attributed to multiple powerful wave generation mechanisms occurring at the scale of individual storms. 408 One primary mechanism is thermal forcing, where the pulsatile nature of latent heat release in a convective updraft acts like a 409 piston on the surrounding stable air, generating a broad spectrum of gravity waves (Beres et al., 2005). A second, 410 complementary mechanism is mechanical forcing, where the body of the strong updraft itself acts as a physical barrier to the background wind. The flow forced over this "moving mountain" generates large-amplitude, low-phase-speed waves that are 411 stationary relative to the storm (Corcos et al., 2025; Wright et al., 2023). The strong spatial correlation shown in Figure 3a 412 413 between the most intense kinetic energy observed by Aeolus and the lowest OLR values (

Figure 4. Spatiotemporal distribution of the Ek/Ep ratio in the ERA5 reanalysis for the equatorial band (10°S-10°N). The UTLS altitudes are defined between one kilometer below the tropopause and 22 km. White and black contour lines represent 210 and 220 W/m2 OLR, respectively. Each bin corresponds to an average of over 3 weeks and 10 degrees. The OLR measurements were obtained from the Australian Bureau of Meteorology. The tropopause is determined from the ERA5 reanalysis.

To illustrate this complexity within a self-consistent framework, we first examine the Ek/Ep ratio derived entirely from the ERA5 reanalysis. Figure 4 presents the longitudinal and temporal variations of this ratio in the equatorial UTLS. The figure immediately reveals that the ratio is far from constant. It exhibits significant spatial and temporal variability, with values frequently exceeding the linear theory predictions (>2). Notably, distinct hotspots of high Ek/Ep ratios are present, particularly over the Indian Ocean at around 70° and the South American continent at 300°.

454 In regions outside of the most intense convective activity, where ERA5 does manage to represent some kinetic energy enhancement, the agreement with Aeolus is often satisfactory. This is visible in Fig. 3c, where the same areas (120° and 300°) 455 456 show a correct correspondence. This suggests that when wave generation is not dominated by deep convection, ERA5 can 457 reproduce realistic GW structures. The strong agreement in these non-convective regions also reinforces the idea that the 458 dominant winds have a strong zonal component, as the quasi-zonal  $u_{HLOS}$  measurement from Aeolus is sufficient to capture 459 these features. The elongated white stripe during February-March 2020 comes from an intense intraseasonal disturbance, the 460 2020 Madden-Julian Oscillation (MJO), which can inject unusually strong gravity-wave energy into the upper troposphere 461 (Kumari et al., 2021).

The divergence between ERA5 and Aeolus becomes most pronounced precisely in the deep convective regions where Aeolus observes its strongest Ek signals, inside the areas of low OLR. The discrepancy appears specifically linked to convectiondriven dynamics, which are either not properly represented or fail to trigger sufficient wave activity within the ERA5 model's parameterizations. This suggests that the primary cause of ERA5's underestimation is not a simple mispartitioning between the horizontal wind components (i.e., a directional bias in the line-of-sight projection) but rather a more fundamental, largescale underestimation of the total kinetic energy.

465

This model-internal result demonstrates that relying on a fixed ratio to infer one energy component from another is likely to 469 be inaccurate, especially in convectively active regions. The partitioning of energy between kinetic and potential forms is itself 470 a key diagnostic of wave dynamics that requires further observational constraints.

One promising possibility of this study in providing deeper context lies in comparing the kinetic energy of gravity waves 472 observed by Aeolus with the potential energy derived from GNSS-RO data. GNSS-RO provides high-resolution temperature 473 profiles that are used to estimate the potential energy of gravity waves. Previous studies that looked into GW climatology all 474 relied on these estimate to base their observations on, as it was the only global instrumentation available (Schmidt et al., 2008;

Alexander et al., 2008b; Sácha et al., 2014; Khaykin et al., 2015). Hence, we will adopt this method of comparison as well.

Figure 5. Comparison between  $Ek_{Aeolus\,HLOS^*}$  (left column) and Ep GNSSRO (right column). Each row corresponds to a season, from June-July-August 2019 to March-April-May 2020. The UTLS altitudes are defined between one kilometer below the tropopause and 22 km. The tropopause is determined from the ERA5 reanalysis. The maps are smoothed using a combination of median and moving average filters as described in the Methods section.

Fig.5 offers a side-by-side seasonal comparison of Ek<sub>Aeolus HLOS\*</sub> (left column) and Ep derived from GNSS-RO (right column), covering the period from June 2019 to May 2020. The figure highlights key spatial and temporal patterns of gravity wave activity detected by each instrument, with both datasets presenting clear seasonal variability.

Although the ratios between Ek and Ep suggested by linear gravity wave theory generally range between 5/3 and 2.0, empirical observations show significant variability. This variability, which is influenced by geographical factors, nonlinear processes, or wave interactions, underscores the importance of examining these two forms of energy from different perspectives rather than seeking strict correspondences.

With that in mind, what stands out from this comparison is the overall consistency in detecting gravity wave hotspots, particularly within the tropical belt (The African land convection and Indian Ocean hotspot are consistent on both instruments). One notable aspect of the comparison is the seasonal shift in gravity wave activity between the two datasets, with both detecting enhanced wave activity during certain months (increased activity levels in DJF and MAM 2020). Because of inherent differences (different line of sight and signal projection, different physical quantities and their varying ratio that is empirically challenging the literature, different signal treatment and correction), direct one-to-one comparisons are not appropriate. Nonetheless, it allows us to draw parallels with Aeolus observations, where spatial and temporal correlation of hotspots should

follow the same disposition, allowing for an independent benchmark. Despite these methodological differences, both instruments align on the seasonal peaks and general distribution of wave activity, reinforcing the reliability of the data.

Figure 6. (a,b,c) Hovmoller diagram of  $Ep_{GNSS-RO}$ ,  $Ep_{ERA5}$  and their difference. White, black and red contour lines represent 210, 220 and 265 W/m2 OLR, respectively. Each bin corresponds to an average of over 3 weeks and 10 degrees. The OLR measurements were obtained from the Australian Bureau of Meteorology. The UTLS altitudes are defined between one kilometer below the tropopause and 22 km. The tropopause is determined from the ERA5 reanalysis. Black stippling indicates regions where the difference between quantities is statistically significant (two-sample t-test, p < 0.05).

The Ep<sub>GNSS-RO</sub> shown in Fig.6a does not closely follow the patterns of OLR activity. As the method employed removes most traces of kelvin waves in the signal, the remaining activity is mostly comprised of GWs. This suggests that Ep does not effectively capture GW activity in regions of deep convection, as indicated by the lowest OLR values. However, it is found that the non-convective areas are seen both on instances of Ek and Ep, in Fig.3a and Fig.6a (with notable examples such as August 2020 around 100°E, as well as in May 2021 and 2022 near 50°E). This observation supports the notion that, in terms of GW activity, deep convective phenomena primarily generate Ek, while less intense convective events (indicated in Fig.6a as occurring in the neighbouring region outside the white contours) produce a more balanced distribution between both energy components. It would be incorrect to assume that no wave activity occurs in low OLR regions; previous studies have shown that Ep values peak at 15 km altitude around the maritime continent, where the Walker circulation rises under non-El Niño conditions (Ern et al., 2004; Yang et al., 2021).

Nonetheless, the Eperals diagram shown in Fig.6b is generally consistent with the results shown in Fig.6a, if one admits that 519 520 the instrumental signal is prone to more noise and higher average values. Particularly in regions outside the primary convection 521 hotspots, in August 2020 around 100°E, we see coherent signals in both datasets. Similarly, in May 2021 near 50°E or in 522 February 2022 near 120°E, distinct patterns emerge in both datasets. These alignments indicate that when gravity waves have 523 a stronger potential energy component, both datasets capture these features, even outside the primary zones of low OLR. It 524 can also be noted that the patterns visible in Fig.6b strongly resemble the patterns presented by ERA5 in Fig.3b, a sign of ERA5's tendency to rely on the existence of Ep to determine the presence of Ek.

527

533

The differences between ERA5 and GNSS-RO data, depicted in Fig. 6c, show a mean absolute difference of 1.96 J/kg. This reflects a reasonable agreement, given that ERA5 assimilates GNSS-RO measurements. While there is a slight positive mean bias of 1.68 J/kg (GNSS-RO > ERA5), which accounts for the prevalence of light red colors in the plot, the differences are scattered and show no large-scale, systematic pattern correlated with convection. This stands in stark contrast to the systematic and large discrepancies observed in the kinetic energy fields.

The Ek differences are not only larger in magnitude, with a standard deviation nearly twice that of Ep (3.16 J/kg vs. 1.82 J/kg) and a maximum underestimation by ERA5 that is almost three times greater (>24 J/kg vs. ~9 J/kg for Ep), but they are also structurally different. The Ek difference plot is dominated by large, cohesive regions of statistically significant positive values (red), indicating a systematic underestimation by ERA5. While some areas do show a negative difference (blue color), these are of small magnitude and, as confirmed by the lack of stippling, are not statistically significant. Most importantly, the peak underestimation of Ek is systematically co-located with the deepest convective regions (inside the low OLR contours), whereas the minor differences in Ep show no such alignment. Taken together, this evidence points to a specific limitation in the reanalysis: the issue is not a general failure to represent wave energy, but a targeted inability of the model's physics and data assimilation system to generate the intense, localized kinetic component of gravity waves originating from strong convection in data-sparse regions.

542 543

An alternative display of Fig.6c as a ratio, along with an F-test, can be found in Appendix E.

# Aeolus Ek / GNSS-RO Ep | UT/LS | 10S - 10N

Figure 7. Relationship between  $Ek_{Aeolus\ HLOS*}$  and Ep from GNSSRO. The UTLS altitudes are defined between one kilometer below the tropopause and 22 km. The tropopause is determined from the ERA5 reanalysis. The tropopause is determined from the ERA5 reanalysis. Black stippling indicates regions where the difference between quantities is statistically significant (F-test, p < 0.05).

Fig. 7 presents the first observationally-derived long-term study of the Ek/Ep ratio, comparing Aeolus's HLOS Ek and GNSS-RO-derived Ep. It illustrates the longitudinal and temporal variations of the Ek/Ep ratio across the equatorial band (10°S to 10°N) from June 2019 to October 2022.

The regions with the highest ratio values are systematically co-located with areas of deep convection, as indicated by the low OLR contours. This is particularly evident over the Indian Ocean (e.g., September-June 2019/20, 2020/21, and 2021/20) and

over the Western Pacific. This observation suggests that, in areas with similar seasonal characteristics, gravity waves tend to transport more kinetic energy during convective events, which amplifies their influence on the overall energy dynamics. The periodic patterns observed in the data also hint at a seasonal component previously observed by Zhang et al. (2010), potentially tied to atmospheric phenomena such as the shifting ITCZ or changes in jet stream dynamics (Hei et al., 2008). These seasonal fluctuations in the Ek/Ep ratio further reinforce the notion that gravity wave behavior is not static but is influenced by broader atmospheric cycles (Ern et al., 2018; Zhang et al., 2010), contrary to the traditional linear theory paradigm in the literature. Statistical significance testing (represented by the black stippling) confirms that these hotspots of high, convection-linked ratios are statistically significant features rather than artifacts.

A significant division between the Indian Ocean and the eastern Pacific, marked by a contrast around 200° longitude, can be noted in both Fig.7 and Fig.4. This contrast reflects underlying geographic factors, including the distribution of large land masses and convective activity. These two factors play a role in the generation and propagation of gravity waves, causing the distinct variations in the ratio between the two energies.

This observational result stands in contrast to the picture presented by the ERA5 reanalysis in Fig.4. While ERA5 also shows variability in its Ek/Ep ratio, its regions of highest ratio are often located outside the main convective centers. This suggests that ERA5 misrepresents the physical link between deep convection and the partitioning of wave energy.

Given that ERA5 successfully assimilates GNSS-RO measurements, specifically bending angles which contain temperature information (and thus has a reasonable representation of Ep), this discrepancy points to a fundamental difficulty in the reanalysis's ability to generate the corresponding kinetic energy component in the right locations. Without direct wind profile assimilation in these data-sparse convective regions, the model's parameterizations and background error covariances fail to create the intense, localized kinetic energy associated with convective gravity waves.

However, it is noteworthy that in some specific regions and periods, such as over the Indian Ocean between June and September of 2019, or in the longitude band around 300°E, a degree of correspondence between the model and observations can be found. This suggests that for certain regimes, the reanalysis can approximate the energy partitioning, but it fails systematically in the most intensely convective areas. These findings reinforce that direct kinetic energy measurements, as provided by Aeolus, are essential for correcting model biases and improving our understanding of the gravity wave energy budget.

# 5. Discussion

Overall, the results presented in this study allow us to discuss and address two main questions. The first consistent observation made, was that ERA5 underestimates Ek distribution in such regions compared to the Aeolus-derived energy, particularly over the Indian Ocean, where conventional radiosonde wind measurements are very sparse. That difference raised questions on the potential reason for such discrepancies: Is this result an overestimation of Aeolus, due to its known increased noise and decaying performance during its life-cycle, or an underestimation for ERA5, due to the lack of direct wind observations assimilated?

The analysis of ALADIN wind profiling and ECMWF ERA5 reanalysis data, provided in Fig.2 and Fig.3, revealed enhanced GW activity over the Indian Ocean during Boreal Summer, as well as over the western Pacific and maritime continent in Boreal Winter. The migration of this enhanced GW activity from eastern Africa to the Pacific maritime continent follows a clear seasonal cycle, strongly linked to deep convection as shown by the correlation with regional OLR minima. This robust seasonal pattern indicates that the underlying wave sources are organized by planetary-scale phenomena, primarily the major tropical monsoon systems (Wright and Gille, 2011). The structures observed by Aeolus are therefore highly consistent with the kinetic energy signature of gravity waves generated by the powerful thermal and mechanical forcing mechanisms (Beres et al., 2005; Corcos et al., 2025) known to occur within the large, organized convective systems of the Asian, African, and Maritime Continent monsoons (Kang et al., 2017; Liu et al., 2022). Previous satellite climatologies have firmly established these monsoon regions as dominant global hotspots for stratospheric gravity wave activity (Hindley et al., 2020; Wright and Gille, 2011). This suggests that Aeolus is effectively capturing these seasonally-driven, convection-induced GWs that are underrepresented in ERA5. One of the persistent features observed throughout the study was the high-energy gravity wave hotspot over the African continent, which remained consistent across seasons and years. This suggests a continuous mechanism of continental convection driving gravity wave activity in this region.

Having established that the Aeolus kinetic energy signal is robust and represents vertically propagating stratospheric gravity waves rather than tropospheric artifacts (as confirmed by our sensitivity analysis in Sect. 2.2), we can use external information to arbitrate the cause of the discrepancy with ERA5. An additional tool at our disposal to solve the case is the global distribution of Ep, through the use of independent GNSS-RO instruments. Our analysis confirms that the assimilation of GNSS-RO data in ERA5 is highly effective, with minimal discrepancies observed between the reanalysis Ep and direct GNSS-RO observations (Fig.6c). This key finding allows us to arbitrate between two potential causes for the Ek discrepancy: a lack of direct wind data assimilation versus inherent biases in the model's physics (e.g., its GWD parameterization).

Several lines of evidence from our study point towards the lack of wind assimilation as the dominant cause. Firstly, the fact that ERA5 accurately reproduces Ep fields demonstrates that the underlying model can represent the thermodynamic signatures of wave activity when properly constrained. Conversely, the largest discrepancies are found in kinetic energy, a purely wind-based quantity, and are concentrated over data-sparse regions like the Indian Ocean, precisely where Aeolus provides direct wind profile measurements not available from other observing systems (Banyard et al., 2021).

Secondly, while ERA5's non-orographic GWD scheme has known limitations and is not directly forced by diagnosed convection (Orr et al., 2010), it is unlikely to be the sole reason for the missing Ek. Such a parameterization bias would be expected to manifest as a systematic error across different variables or regions, or as a persistent model drift requiring large, ongoing corrections by the assimilation system (Dee, 2005). However, our findings show a targeted deficiency: the model performs well on assimilated temperature (Ep) but poorly on unassimilated wind (Ek) in the very same locations. This sharp contrast strongly suggests the problem is not a wholesale failure of the model's physics to generate wave energy, but rather its inability to correctly partition that energy into kinetic and potential components without direct wind constraints.

In data-sparse areas, ERA5 must rely on its internal background error covariances to infer wind adjustments from the assimilated mass field (Hersbach et al., 2020). These statistical relationships are primarily designed to represent large-scale, quasi-balanced (rotational) flow and have long been known to be less effective at specifying the smaller-scale, divergent component of the wind field to which convectively generated gravity waves belong, especially in the tropics (Žagar et al., 2004). While the Integrated Forecasting System (IFS) has evolved considerably, recent Observing System Experiments (OSEs) using Aeolus data confirm that this challenge persists. These studies provide direct evidence that the assimilation of Aeolus wind profiles systematically enhances the analyzed amplitudes of equatorial waves, particularly in regions of strong vertical wind shear where the model's background state is most uncertain (Žagar et al., 2021, 2025). Consequently, while the assimilation of GNSS-RO constraints the thermodynamic (Ep) aspect of the wave, the system lacks the necessary information and dynamic constraints to generate the corresponding divergent wind perturbations, leading to the observed Ek deficit. This process evidently fails to capture the full spectrum of high-Ek wave modes generated by convection.

Overall, the findings presented here are in full agreement with the elements outlined in the introduction, suggesting that ERA5 is underestimating the Ek component. Indeed, ERA5 has several known shortcomings, such as its underrepresentation of eastward-propagating inertio-gravity waves (Bramberger et al., 2022), its site-dependent errors in tropical regions (Campos et al., 2022), and the broader limitations of data assimilation systems in capturing circulation dynamics, particularly in areas with sparse wind observations (Podglajen et al., 2014; Žagar et al., 2004). These challenges are particularly evident in the representation of key tropical phenomena like the Quasi-Biennial Oscillation (QBO), which is driven by the upward propagation and dissipation of a spectrum of atmospheric waves. Recent studies using direct Aeolus wind observations have provided new insights into how reanalyses represent these processes. For instance, Banyard et al. (2023) found that during the 2019/2020 QBO disruption, a period covered by our study, the onset of the disruptive easterly jet was observed by Aeolus five days earlier than in ERA5. This discrepancy was linked to higher Kelvin wave variances and sharper vertical wind shear in the Aeolus data, suggesting that ERA5 may misrepresent the breaking of smaller-scale waves that are crucial for forcing the QBO.

Similarly, Ern et al. (2023) confirmed that while the zonal-mean QBO is well-represented in ERA5, local biases exist, 644 645 particularly in shear zones. From a data assimilation perspective, Žagar et al. (2025) showed that assimilating Aeolus winds 646 produced the largest changes to the analyzed state in the UTLS precisely during the 2019/2020 QBO disruption, highlighting 647 the importance of direct wind observations for reducing uncertainties in these critical shear zones. Together, these findings, 648 derived from the same novel wind dataset used here, support our conclusion that reanalyses can have significant deficiencies 649 in representing the full spectrum of wave activity and its associated kinetic energy in the absence of direct wind assimilation.

Another discussion enabled by Aeolus observations concerns the longstanding assumption of a constant Ek/Ep ratio in GW studies. Specifically, the question arises: Is the conventional view of a constant ratio for inferring Ep from Ek (and vice versa) still tenable? Or do the new data suggest that this ratio is no longer universally valid in real-world, often non-linear, atmospheric conditions?

At first glance, using a fixed ratio appears straightforward for converting well-documented Ep (from temperature-based 655 instruments such as GNSS-RO) to Ek. Traditionally, linear GW theory proposes a near-constant ratio of Ek to Ep, often quoted 656 between 5/3 and 2.0 (VanZandt, 1985; Hei et al., 2008). In idealized models of linear wave behavior, the kinetic and potential 657 energies are expected to be comparable, leading to a ratio close to unity. This theoretical relationship has been confirmed 658 observationally. In stable, linear wave conditions, the energy ratios adhere closely to predictions (Nastrom et al., 2000), a 659 finding supported by a modern case study of individual, freely-propagating waves (Huang et al., 2021).

However, a growing body of evidence challenges this simplification: Empirical work increasingly reveals significant variability in this ratio, indicating non-linear effects in real-world atmospheric conditions (Wing et al., 2025; Baumgarten et al., 2015; Guharay et al., 2010; Tsuda et al., 2004). When the observed energy ratios deviate significantly from this expected range, non-linear processes may be at play. While a large climatological study may find a mean Ek/Ep ratio close to theoretical values (e.g., 1.5 in Zhang et al., 2022), this average can mask significant event-to-event variability. For instance, in situations where wave amplitudes are particularly large, wave-wave interactions, such as those resulting from wave breaking or saturation, could lead to the observed discrepancies. This has been demonstrated in earlier work by Mack and Jay. (1967), who found that under certain conditions, potential energy deviated markedly from kinetic energy, suggesting non-linear effects. Similar findings have been reported by Fritts et al. (2009), who showed that interactions between gravity waves and fine atmospheric structures can result in turbulence, thereby affecting the balance between kinetic and potential energy. A recent study also confirmed that the ratio is not static and can be actively modulated by the background atmospheric state, such as strong wind shear (Wing et al., 2025).

With everything in place to link these elements, the observed comparison in Fig.4 of the Ek/Ep ratios from ERA5, Aeolus, and GNSS-RO confirms that the characteristics of gravity waves vary significantly across time and space. The observed ratios, 1.7 (+/- 0.38) for ERA5, 1.4 (+/- 0.54) for Aeolus/GNSS-RO, indicate that the waves encompass both linear and non-linear processes. The frequent observation of ratios exceeding unity, aligning with trends identified in previous studies, suggests that a substantial portion of the waves' energy is contained in kinetic form, often indicative of non-linear behavior. Because the assumption of a constant ratio is increasingly challenged by empirical observations (see references in the previous paragraph), it accentuates the need to shift the paradigm from relying solely on temperature perturbations to directly deriving Ek. As such, directly measuring kinetic energy is a major missing link for a comprehensive understanding of GW dynamics.

Beyond these considerations of gravity wave dynamics and energy ratios, we should also acknowledge the limitations of the Aeolus satellite. These include both its technical shortcomings and the constraints imposed by its HLOS projection, which directly impact the representativeness of its measurements. A 1.6 ratio was determined for Ek/Ek<sub>HLOS</sub> using ERA5 (as detailed in Sect. 2.2 and shown in Appendix C). It reflects the efficiency with which HLOS winds from Aeolus can approximate the full kinetic energy field. The ratio indicates that HLOS winds account for approximately 62.5% of the total Ek, while the remaining 37.5% is undetectable due to the projection limitations of HLOS measurements. The discrepancy suggests that the HLOS winds alone cannot fully capture the energy contributions from multi-dimensional wave dynamics. However, this ratio can help estimate the full Ek indirectly with reasonable accuracy. While this approach introduces some assumptions, it can be further refined by cross-validating against comprehensive datasets from reanalyses like ERA5.

Understanding the vertical wavelength of convective GWs is an essential element for characterizing their dynamics. However, Aeolus is inherently limited in retrieving accurate vertical wavelengths due to its design. The placement of range bins was fixed at the time of observation, introducing inconsistencies in vertical resolution that affect the precise identification of wave peaks and troughs. Additionally, the N/P parameter, which controls the number of accumulated measurements (N) and pulses (P) per cycle, introduces variability in the horizontal resolution of Aeolus data. Changes to this setting, such as the transition from N=30 to N=5, improve horizontal resolution but exacerbate the misrepresentation of vertical wave structures. Furthermore, any spectral analysis of a finite vertical profile is inherently constrained. For geophysical spectra that are typically having more variance at longer wavelengths, a simple peak-finding method would likely identify a dominant wavelength that is an artifact of the analysis window or filtering choices. Given these limitations, we limit our analysis to the vertically-integrated energy within a defined band-pass filter (vertical wavelengths between 1.5 and 9 km), which is a more robust quantity.

Nevertheless, we can speculate that the high Ek values observed by Aeolus in convective regions are associated with shorter-wavelength waves. This interpretation is consistent with established physical mechanisms which state that waves with high EK are typically generated in regions with strong convective updrafts and downdrafts, where the rapid vertical movement of air masses creates intense small-scale disturbances. These localized and transient disturbances, arising from geostrophic imbalance, generate GWs that carry energy away from the convective region, where strong forcing efficiently transfers energy into the EK spectrum at shorter wavelengths (Waite and Snyder, 2009). The correlation between high EK and shorter

wavelengths is particularly pronounced in convective systems, as confirmed in both observational and numerical estimations (Kalisch et al., 2016), especially in tropical regions and cyclones (Chane Ming et al., 2014). A definitive observational confirmation of this from the satellite itself, however, remains a challenge due to the aforementioned limitations.

Looking forward, a critical application for such observations is the constraint of gravity wave momentum fluxes, which are essential for global circulation models. However, deriving momentum flux estimates directly from singlecomponent wind measurements like those from Aeolus presents two co-dependent problems. First, the vertical flux of horizontal momentum (e.g., (u'w')) requires simultaneous knowledge of horizontal (u') and vertical (w') wind perturbations. Aeolus supplies only the line-of-sight projection of the horizontal wind and, crucially, no direct information on the vertical wind. In the standard processing w' is simply assumed negligible (Krisch et al., 2022), leaving the key term in the flux equation unconstrained. Second, the satellite's sampling geometry further limits what can be inferred. Aeolus observes with a ~3 kmwide "pencil beam" that is horizontally averaged to about 86 km along track, and its sun-synchronous orbit completes ~16 revolutions per day (roughly 32 equator crossings, or 15–16 every 12 h). Small-scale gravity waves are therefore captured only where the narrow ground tracks happen to intersect them, leaving large spatial and temporal gaps. Together, the absence of direct w' measurements and this sparse, one-dimensional sampling mean that Aeolus winds alone cannot yield global momentum-flux maps without substantial modelling support or complementary observations.

A potential pathway to overcome this limitation involves creating synergistic datasets, for instance by combining Aeolus wind data with simultaneous, collocated temperature measurements from instruments like GNSS-RO. In principle, gravity wave polarization relations could then be used to infer the missing wind components. However, this approach is not a simple remedy and relies on strong, often unverifiable, assumptions about unmeasured wave parameters, including the horizontal wavelength, intrinsic frequency, and the stationarity of the wave field between measurements (Alexander et al., 2008a; Chen et al., 2022).

Therefore, while Aeolus does not directly measure momentum flux, its unprecedented global measurements of kinetic energy provide an additional observational constraint. Such observations are a critical prerequisite for developing and testing the more complex, multi-instrument techniques that will be required to eventually constrain the global gravity wave momentum budget.

#### 6. Conclusion

712

- In this study, we examined the capacity of the Aeolus ALADIN instrument to capture and resolve GWs in tropical UTLS. 732 While this task might appear challenging at first, because of the data alteration issues Aeolus faced during its lifecycle, the 733 study proposed a noise correction process, which used ERA5 reanalysis as a reference to estimate and correct for Aeolus's
- instrument-induced variance. This correction improved the retrieving of kinetic energy, and our comparison with collocated

- radiosonde data further validated that approach. A key focus of our analysis was the ratio between kinetic and potential energies
- (Ek/Ep), providing insights into the linear or non-linear nature of these waves.
- The principal findings can be summarized as follows:

- Aeolus observations capture significant kinetic energy enhancements over tropical convection hotspots, particularly over the Indian Ocean, where ERA5 shows substantial underrepresentation due to sparse wind observations.
- Direct wind data from Aeolus could significantly enhance tropical UTLS reanalysis products, particularly in convection-driven GW regimes, reducing biases in Ek representation.
- In many regions with strong convective forcing, Aeolus data suggest a larger kinetic energy component, pointing to wave breaking, saturation, and other non-linear processes that depart from purely linear wave dynamics.
- While linear GW theory often prescribes an Ek/Ep ratio between ~1.6 and ~2.0, our results show that this ratio can vary significantly, depending on location and season. This highlights the need for direct kinetic-energy measurements rather than relying solely on temperature-derived potential energy as a proxy.
- Aeolus also helps fill this gap. However, given its HLOS projection, Aeolus underestimates the total Ek if meridional components are significant, reinforcing that multi-instrument approaches are mandatory for accurately characterizing GW fields.

Thus, this study has demonstrated the value of Aeolus Rayleigh wind profiling for observing GWs in the tropical UTLS, despite the high and time-variable random error associated with its measurements. Our findings confirm that the annual and zonal variation of GW activity in the tropical tropopause layer and lower stratosphere is modulated by deep convection, as demonstrated by Dzambo et al. (2019) and Evan et al. (2020). Furthermore, Aeolus data expose a significant need for improving the reanalysis regarding the convective GW Ek. The lack of GW-derived Ek in ERA5 is most pronounced in the Indian Ocean region, where conventional radiosonde wind measurements are relatively sparse. It is highly likely that the missing Ek in ERA5 is due to the misrepresentation of convective processes. The results also indicate that standard assumptions about the Ek/Ep ratio do not always hold, particularly under convective or otherwise non-linear conditions. Aeolus' range-bin design and horizontal integration restrict its ability to determine wavelengths with accuracy, which poses a significant challenge for fully capturing the characteristics of GW. This limitation highlights the need for complementary datasets, which could be addressed in newer iterations of the instrument. While this study delivers some insights into UTLS GW activity and the benefits of global wind observation, future research should continue investigating the factors contributing to the

discrepancies observed between Aeolus and ERA5 data. The kinetic energy constraints provided here represent a novel step, and future missions like Aeolus-2 will be essential for developing the synergistic techniques required to ultimately quantify the global momentum transport by these waves.

# **Appendix A: Sensitivity Analysis**

Figure A1. Sensitivity of Aeolus profile counts per analysis bin to the choice of spatio-temporal averaging domain.

Figure A1 presents the histograms of the number of Aeolus profiles available per spatio-temporal bin for four different averaging domain configurations: (a) the baseline 20°x5°, 7-day domain used in this study, (b) a spatially finer 10°x2.5°, 7-day domain, and (c) a temporally finer 20°x5°, 3-day domain. Red dashed lines indicate the median profile count for each configuration. Panel (d) shows the cumulative distribution function for all four configurations, including a domain that is finer in both space and time. The results demonstrate the trade-off between domain size and sampling density. While finer domains offer higher resolution, they significantly reduce the number of profiles available for robustly calculating the background state, with a majority of bins containing fewer than 20 profiles. The baseline configuration (blue line) was chosen as it ensures a

sufficient number of profiles per bin (median of 71) to minimize noise and provide a stable background estimate, as discussed in the main text.

# Appendix B: Aeolus sensitivity to the tropopause selection

Figure B1. Scatter plot comparison of Aeolus-derived kinetic energy (Ek) calculated using different vertical averaging layers.

Figure B1 tests the sensitivity of the derived kinetic energy to the vertical averaging layer, addressing the potential contamination from upper-tropospheric non-wave outflow. The x-axis on both panels represents the Ek calculated using our original layer (starting 1 km below the tropopause). (a) The y-axis shows Ek calculated using a "Conservative" layer starting 1 km above the tropopause. (b) The y-axis shows Ek from a "Very Conservative" layer starting 2 km above the tropopause. The strong linear correlation (r = 0.92 and r = 0.83, respectively) and the fact that the data points cluster near the 1:1 line (red dashed line) demonstrate that the majority of the energy signal is retained when the upper troposphere is excluded. The linear fit (blue solid line) shows a predictable reduction in magnitude but confirms that the underlying spatial patterns of high and low energy are highly consistent across all layers.

Figure B2. Zonal mean structure of time-averaged Aeolus kinetic energy (Ek) for different vertical averaging layers.

Figure B2 shows the time-averaged, zonal mean Ek as a function of longitude for the three different vertical averaging layers defined in Figure B1. The black line represents our original layer, the blue dashed line is the "Conservative" layer (+1 km), and the red dotted line is the "Very Conservative" layer (+2 km). The plot confirms that while the absolute magnitude of Ek decreases as the layer is moved higher into the stratosphere (as expected due to wave dissipation), the geographical structure of the energy hotspots is remarkably stable. The primary peaks of high energy (e.g., over the African/Indian Ocean sector from ~60-120°E and the Americas/Atlantic sector from ~280-320°E) persist across all three calculations. This provides strong evidence that the observed energy hotspots are robust features originating from vertically propagating gravity waves that have reached the lower stratosphere, and not artifacts of shallow upper-tropospheric outflow.

# Appendix C: EK / EK HLOS Ratio

Figure C1. Temporal and spatial variability of the ratio between  $E_{k\,HLOS}$  and Ek in the ERA5 model over the tropical region (30°S–30°N) for the year 2021

Figure C1 displays the ratio between  $E_{k\,HLOS}$  and Ek in the ERA5 model, over the tropical region for the UTLS. The ratio values range from 1.5 to 3, between January 2021 and December 2021 included, depicting variations in how well the HLOS measurements capture the total kinetic energy in this region.

This map illustrates the anisotropy of the gravity wave field as represented in the ERA5 model for the year 2021, chosen for its significant wave activity which highlights the regional patterns of energy partitioning. The ratio quantifies the contribution of the unobserved meridional wind component to the total kinetic energy. A ratio of 1 (blue colors) indicates purely zonal wave energy, while higher values (red colors) signify an increasing contribution from meridional motions.

A distinct pattern emerges along the equatorial belt (10°S - 10°N). Over a vast longitudinal sector stretching from Africa, across the Indian Ocean, and over the Maritime Continent (approximately 0°E to 160°E), the ratio remains low, generally below 2.0. This indicates that wave energy is predominantly zonal in these regions, meaning a quasi-zonal HLOS measurement like that from Aeolus is expected to capture a large fraction of the total kinetic energy. This aligns with the influence of persistent large-scale zonal flows in this sector, such as easterly trade winds.

In contrast, the sector from approximately 160°E to 280°E, encompassing the central and eastern Pacific, shows significantly higher ratios, often exceeding 2.5. This points to a wave field with a much stronger meridional component, where an HLOS measurement would systematically underestimate the total kinetic energy. A third regime is observed over the Atlantic and South America (from 280°E to 360°E), where the ratio is more variable but consistently elevated, indicating a mixed wave field with a significant, though not always dominant, meridional energy component.

# Appendix D: Aeolus Instrumental Noise Correction Methodology

849

This appendix provides a detailed, step-by-step description of the spatio-temporally adaptive algorithm used to correct for time-varying instrumental noise in the Aeolus-derived gravity wave (GW) kinetic energy (Ek) data. The objective is to produce a corrected dataset,  $\overline{Ek_{Aeolus\,HLOS*}}$ , that removes instrumental bias while preserving the unique, high-fidelity geophysical signals of GWs that Aeolus observes. The correction methodology involves two primary stages: first, a detailed correction applied to time-longitude Hovmöller diagrams, and second, a refinement of this correction for application to broader geographical maps.

### Part 1: Correction of Time-Longitude Hovmöller Diagrams

- As introduced in the main text, our best estimate of the noise energy is denoted as  $\overline{Ek_{\text{L.N.}}}$ . Within this appendix, where specific spatio-temporal and component estimates of this noise are derived, we will use the "hat" notation to explicitly denote these estimated noise values, such as  $\widehat{Ek_{\text{L.N.}}}$  for the time-longitude noise estimate. The initial estimation of the noise term is performed on the 2D Hovmöller data matrices of Aeolus  $E_{k,\text{Aeolus}HLOS}(t,\phi)$  and ERA5  $E_{k,\text{ERA5}HLOS}(t,\phi)$  where t represents time steps (3-week averages) and  $\phi$  represents longitude bins. This correction is conducted within the 10°S–10°N latitude band where deep tropical convection is most prominent. The algorithm adaptively estimates the instrumental noise component  $\widehat{Ek_{\text{L.N.}}}(t,\phi)$  which is then subtracted from the observed Aeolus energy as shown conceptually in Eq. (6) of the main text.
- Step 1: Defining an Activity-Based Blending Weight
- The core of the adaptive algorithm is a blending weight,  $W(t, \phi)$ , that determines the balance between an additive and a multiplicative correction approach. This weight is derived from the ERA5 Ek, which serves as a proxy for the true level of atmospheric GW activity. First, a background reference energy  $E_{k,bg}$  (bg standing for background) defined as the median ERA5 Ek within a pre-defined quiescent reference sector (200°E–250°E over the Pacific Ocean), chosen for its typically low convective activity. An activity index, x(t,L), is then computed for each point in the Hovmöller diagram:

$$x(t, \phi) = \frac{E_{k,ERA5HLOS}(t, \phi) - E_{k,bg}}{E_{k,bg}}$$
 (7)

From this index, a logistic function is used to create a preliminary activity map, A(t,L), ranging from 0 (quiescent) to 1 (active):

$$A(t, \phi) = \frac{1}{1 + e^{-x(t, \phi)}}$$
 (8)

To ensure spatial coherence and prevent abrupt transitions, this activity map is smoothed using a 2D Gaussian filter with a standard deviation of 2 pixels in the time dimension and 10 pixels in the longitude dimension. The final background-confidence weight, W(t,L), is then defined as:

$$W(t, \phi) = 1 - A_{smoothed}(t, \phi) \tag{9}$$

Thus, W is close to 1 in quiescent regions and approaches 0 in highly active regions.

### 856 Step 2: Deriving the Additive and Multiplicative Correction Components

- Two candidate correction components are calculated. The additive component is designed to correct for the baseline
- instrumental noise offset. A time-dependent offset vector,  $\delta_{noise}(t)$ , is computed by taking the median difference between
- Aeolus and ERA5 Ek within the same quiet reference sector used in Step 1:

$$\delta_{noise}(t) = \text{median}_{\phi \in \text{ref. sector}} \left( E_{k,AeolusHLOS}(t, \phi) - E_{k,ERA5HLOS}(t, \phi) \right)$$
 (10)

- This offset vector, which captures the time-varying instrumental bias trend, is then smoothed temporally with a 7-point (21-
- week) moving median to reduce noise. The additively corrected energy field,  $E_{k,bias-adj}$  is then defined as:

$$E_{k,bias-adj}(t,\phi) = E_{k,AeolusHLOS}(t,\phi) - \beta \cdot \delta_{noise}(t)$$
 (11)

- where  $\beta$  is an offset relaxation factor (set to 1.15) to empirically fine-tune the subtraction.
- The multiplicative component is designed for active regions where noise effects might scale with the signal. A single, mission-
- period Activity Ratio Scalar,  $\mathcal{R}_{\mathcal{A}/\mathcal{E}}$ , is computed as the median ratio of Aeolus to ERA5 Ek over all grid points where the
- activity weight  $A(t,\phi)$  from Eq. (8) is greater than 0.5:

$$\mathcal{R}_{\mathcal{A}/\mathcal{E}} = \text{median}_{(t,\phi) \in \text{active}} \left( \frac{E_{k,AeolusHLOS}(t,\phi)}{E_{k,ERA5HLOS}(t,\phi)} \right)$$
 (12)

The Ratio-Scaled Energy,  $E_{k,ratio-scaled}$ , is then:

$$E_{k,ratio-scaled}(t, \phi) = \frac{E_{k,AeolusHLOS}(t, \phi)}{\mathcal{R}_{A/F}}$$
 (13)

## 875 Step 3: Blending and Final Hovmöller Correction

- The final corrected Hovmöller field,  $Ek^*(t, \phi)$ , is a weighted average of the two candidates, using the weight map W from Eq.
- (9):

854

864

$$E_k^*(t,\phi) = W(t,\phi) \cdot E_{k,bias-adj}(t,\phi) + [1 - W(t,\phi)] \cdot E_{k,ratio-scaled}(t,\phi)$$
 (14)

Any resulting negative values are clipped to zero. The estimated noise component,  $\overline{Ek_{I.N}}$ , as referenced in Eq. (6) of the main text, is therefore equivalent to the total amount subtracted:

$$\overline{Ek_{LN}} = \overline{Ek_{Aeolus\,HLOS}} - \overline{Ek_{Aeolus\,HLOS*}}$$
 (15)

Figure D1. Breakdown of the Hovmöller diagram correction. (a) Raw Aeolus Ek before correction. (b) The estimated noise correction matrix, Ek,I.N. that is subtracted. Note the increasing trend over time, reflecting instrument degradation, and the spatial structure modulated by the blending of additive and multiplicative components. Stippling indicates regions where the corrected Aeolus Ek is statistically different from ERA5. Ek (F-test, p<0.05). (c) Final corrected Aeolus Ek (Ek,Hovmöller\*). Background energy levels are significantly reduced while convective hotspots are preserved. White and black contour lines represent 210 and 220 W/m2 OLR, respectively.

Figure D1 illustrates the core steps of the spatio-temporally adaptive noise correction. (a) The raw, uncorrected Aeolus Ek Hovmöller diagram, showing both geophysical signals and a clear increasing trend in background noise over the mission lifetime. (b) The estimated noise component derived using the adaptive algorithm. Note the temporal increase reflecting instrument degradation, as well as the spatial structure modulated by the blending of additive and multiplicative corrections. (c) The final corrected Aeolus Ek. The background energy levels have been significantly reduced, removing the artificial trend,

while the physically meaningful convective hotspots are preserved and sharpened. White contours indicate regions of deep convection (OLR < 220 W m<sup>-2</sup>).

#### Part 2: Refinement of Noise Correction for Seasonal Geographical Maps

- For the seasonally averaged geographical maps, which cover a broader latitudinal range (0°S–30°N, denoted by latitude θ)
- than the initial Hovmöller analysis, the noise estimation is refined. For each season (e.g., DJF, MAM, JJA, SON), the estimated
- noise field from the Hovmöller analysis  $\widehat{Ek_{LN}}$ , s averaged over the relevant time steps to yield a Seasonal Mean Longitudinal
- Noise Profile  $\overline{\widehat{Ek}_{LN}}$ . This yields a seasonally representative longitudinal noise profile. This profile forms the basis for
- correcting the uncorrected, seasonally-averaged raw Aeolus Ek map,  $E_{k,Aeolus,map}(\theta,\phi)$ . The refinement proceeds in two steps:
- latitude-weighting and iterative adjustment.

896

#### 903 Step 1: Latitude-Weighting of the Longitudinal Correction

- The impact of instrumental noise may be proportionally larger in regions of higher true GW energy. To approximate this, the
- base longitudinal correction profile is weighted by the latitudinal structure of the raw Aeolus Ek map itself. First, the mean
- raw Aeolus Ek is calculated for each latitude band,  $E_{k,Aeolus,map}$ . This is normalized to create a latitude weight profile,  $W_{\theta}$ :

$$W_{\theta}(\theta) = \frac{\overline{E_{k,Aeolus,map}(\theta)}}{\max(\overline{E_{k,Aeolus,map}(\theta)})}$$
 (16)

- This weight is clipped between 0 and 1. This latitudinal weight is then broadcast across the longitudinal correction profile to
- create an initial 2D correction field for the map,  $\Delta E_{map,initial}$ :

$$\Delta E_{map,initial}(\theta, \phi) = \overline{\widehat{E}}_{k,I.N.}(\phi) \cdot W_{lat}(\theta)$$
 (17)

#### 911 Step 2: Iterative Adjustment to Prevent Negative Kinetic Energy

- A simple subtraction of  $\Delta E_{map,initial}$  from  $E_{k,Aeolus,map}$  could result in non-physical negative Ek values where the
- estimated correction is large relative to the observed Ek. To prevent this, a Columnar Clipping Factor,  $\gamma_{clip}$ , is calculated for
- each longitude column of the map. This factor is defined as the minimum ratio of the observed energy to the initial correction
- estimate across all latitudes in that column, ensuring the correction never exceeds the available energy:

$$\gamma_{clip}(\phi) = \max\left(0, \min\left(1, \min_{\theta}\left[\frac{E_{k,Aeolus,map}(\theta,\phi)}{\Delta E_{map,initial}(\theta,\phi)}\right]\right)\right)$$
 (18)

The final 2D correction field applied to the map is then:

$$\Delta E_{map,final}(\theta, \phi) = \Delta E_{map,initial}(\theta, \phi) \cdot \gamma_{clip}(\phi)$$
 (19)

## 919 Step 3: Final Corrected Map and Smoothing

922

925

926

927

The final corrected seasonal map, denoted as  $\overline{E_{k,AeolusHLOS}^*}$ , is:

$$\overline{E_{k,AeolusHLOS}^*}(\theta, \phi) = E_{k,Aeolus,map}(\theta, \phi) - \Delta E_{map,final}(\theta, \phi)$$
 (20)

Finally, a light 2D spatial smoothing (3x3 moving median followed by a 3x3 moving mean) is applied to the corrected map to reduce pixel-scale noise introduced by the gridding and correction process.

Figure D2. Illustration of the noise correction refinement process for a seasonal geographical map (JJA 2021 shown as an example).

Figure D2 details the refinement steps used to adapt the noise correction for application to 2D seasonal maps. (a) The raw, uncorrected seasonally averaged Aeolus Ek map. (b) The initial noise estimate, created by applying a latitudinal weighting to the seasonal-mean longitudinal noise profile derived from the Hovmöller analysis. (c) The final noise estimate, after an iterative adjustment step that prevents non-physical negative energy values. (d) The final corrected seasonal map, representing the geophysical gravity wave kinetic energy field after the removal of instrumental artifacts. This process ensures that the correction is physically consistent across the entire map domain.

## **Diagnostic Validation of Noise Correction**

Figure D3. Difference between the Radiosonde-derived (black) and ERA5-derived (red) estimated noise correction, resulting in the difference between the uncorrected Aeolus HLOS GW Ek (blue) and the ERA5-corrected Aeolus HLOS GW Ek (green).

Figure D3 provides an independent validation of the estimated noise trend by comparing the zonally-averaged  $\delta_{noise}(t)$ , term from Eq. (10) with noise estimates derived from collocated radiosonde measurements at La Réunion, as in Ratynski et al. (2023). It is intended to demonstrate that such a method of instrumental noise estimation is qualitatively consistent with the classical approach based on collocated reference measurements applied in (Ratynski et al., 2023).

The Météo-France upper-air soundings in La Réunion (Aéroport Gillot) was used for the conduct of this analysis. For each collocated radiosonde profile with an Aeolus overpass (within 200 km and +/- 6 hours), we downsampled the radiosonde profile resolution to be equivalent to ALADIN vertical bins. A point-wise difference is then calculated, and the standard

deviation of these differences is what we refer to as random error. In principle, if Aeolus would not experience any degradation through its systems, this standard deviation would remain stable over the years and periods. However, since we observe an increase, as reported by Ratynski et al. (2023, their Fig.6), a link can be made between the instrument degradation and this increase, wrongly attributing signal-to-noise. Squaring this noise estimation provides a metric homogenous to the observed Ek, representing the repercussions of noise on Ek estimation:

$$\operatorname{Ek}_{\operatorname{LN}} = \frac{1}{2} \left( \sigma_{\operatorname{Aeolus-Radiosondes}} \right)^2$$
 (21)

956

While both methods provide similar trends, the model approach remains the safest estimation when considering the potential biases.

# Aeolus / ERA5 GW Ek | UT/LS | 10S - 10N GNSS-RO / ERA5 Energy Ratio | UT/LS | 10S - 10N

Figure E1. Ratio-based comparison of observed and reanalyzed gravity wave energy for the equatorial band (10°S-10°N), between observational data and ERA5. The UTLS altitudes are defined between one kilometer below the tropopause and 22 km. White and black contour lines represent 210 and 220 W/m2 OLR, respectively. Each bin corresponds to an average of over 3 weeks and 10 degrees. The OLR measurements were obtained from the Australian Bureau of Meteorology. The tropopause is determined from the ERA5 reanalysis. The panels show the base-10 logarithm of the ratio between observational data and ERA5.

Figure E1 provides an alternative, ratio-based view of the comparisons presented in the main text, complementing the difference-based analysis. The color scale represents the base-10 logarithm of the ratio, where positive values (red) indicate that the observational dataset has higher energy than ERA5, and negative values (blue) indicate the opposite.

The first panel reinforces the findings from the main text, showing a systematic and significant overestimation of kinetic energy by Aeolus relative to ERA5 (predominantly red colors) within the convective regions identified by low OLR. The ratio

- frequently exceeds 2 (log<sub>10</sub> ratio > 0.3), particularly over the Indian Ocean and Maritime Continent, confirming that ERA5
- substantially underestimates convection-driven kinetic energy.
- In contrast to the kinetic energy, the potential energy ratio is much closer to unity ( $\log_{10}$  ratio  $\approx 0$ ). The colors are predominantly
- neutral or light shades of blue/red, indicating that ERA5 and GNSS-RO have very similar magnitudes of potential energy. This
- is expected, as ERA5 assimilates GNSS-RO temperature data.
- Taken together, these two panels provide evidence for the central conclusion of this study: the discrepancy between
- observations and reanalysis is specific to the unassimilated kinetic energy component. While ERA5 successfully reproduces
- the potential energy field it is constrained by, it fails to generate the corresponding kinetic energy associated with convection,
- a gap that direct wind observations from Aeolus can fill. Black stippling indicates regions where the ratios are statistically
- significant (F-test, p < 0.05).
- Data Availability: Aeolus data are publicly available through the Aeolus online dissemination system (https://aeolus-
- ds.eo.esa.int/oads/access/). The dataset used for the realization of this study can be found at
- https://doi.org/10.5281/zenodo.8113261
- **Author Contributions:** Conceptualization, M.R., S.K., A.H., and M.J.A.; methodology, M.R., S.K., A.H., and M.J.A.; software,
- 984 M.R. and A.M. .; validation, M.R, S.K, A.H, and M.J.A;—original draft preparation, M.R. and S.K. .; writing—review and
- editing, M.R., S.K., A.H., M.J.A., A.M., P.K and A.M.; funding acquisition, S.K., P.K. and A.M.; All authors have read and agreed
- to the published version of the manuscript.
- Competing interests: The contact author has declared that none of the authors has any competing interests.
- **Acknowledgments:** The work by Mathieu Ratynski was carried out under a PhD fellowship co-funded by CNES and ACRI-
- ST. Additional funding has been provided via the CNES APR Aeolus project. MJA was funded by NSF grants #2110002 and
- #1642644.

982

987

989

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
