# Peer review of "Convection-generated gravity waves in the tropical lower stratosphere from Aeolus wind profiling, GNSS-RO and ERA5 reanalysis"

_EGUsphere, 2025_

## Author Comment (AC1)

Reply to Reviewer #2 :

We thank Reviewer #2 for their careful and candid feedback. The comments were helpful in clarifying aspects of the scientific discussion and in prompting us to address some shortcuts and approximations in the original text. We have revised the manuscript accordingly and provide detailed responses to each point below.

The work described here flows from a unique and novel data source, a wonder of the age, and seems fairly sound. However, the figures are not all they could be, and not fully explained, lacking any statistical analysis or mentions of evident artifacts. Many (incomplete) data handling details and other science issues are sprinkled though a rather smooth vague narrative, with too little scientific circumspection. With a modest amount more care, perhaps aided by senior author input, this paper could become as excellent as the data deserve.

The manuscript suffers from three problems common in the dissertation-to-literature translation, detailed further below but listed thematically here:

1. Key technical details for understanding the results are too buried. In place of crisp exposition and taut circumspection are meandering vague mentions of issues around methods, defensive at times and elsewhere a sales pitch for the arbitrary trade-offs settled on. This may be how a committee explained a complex recipe to a student, but is not ideal for a paper addressing peer researchers.

2. Meandering threads are also present as a means for a student to telegraph to a committee their awareness of a reading list, cited often in vague mentions rather than claim-supporting paraphrases of the content. This may be good for dissertations written to a captive readership, but is not ideal for peer researchers. As just one instance, not all winds on larger scales than the filter belong in a named list of Mode Types: the wind is the wind.

3. Findings and interpretations are a bit vague, unobservant of the actual figure set, and repetitive (for instance the "especially Indian Ocean" trope appearing several times). False color scales are unhelpfully distorting, and questions raised by the figures and results are not pursued with a scientifically committed vigor. This may embody the rationally bounded commitment of a student to the physical problem at hand, but ideally a senior coauthor might exert the leadership to bring more depth of inquiry and perspective.

We acknowledge the reviewer's thematic concerns regarding the clarity of technical details, the narrative style, and the depth of the scientific interpretation. In response, we have performed a substantial rewrite of several key sections. Specifically, Section 2 (Data and Methods) has been reworked to provide a better description of the data processing, free from promotional language. Section 3 and 4 (Results) now feature improved figures with perceptually uniform colormaps and appropriate statistical testing to support our claims with greater scientific rigor. The narrative has been tightened to focus on direct interpretation of the figures. Section 5 (Discussion) has been expanded to address the scientific questions raised by the reviewer with more depth and circumspection, particularly regarding the potential for misinterpretation of convective outflow, the physical mechanisms behind our observations, and the limitations of the analysis.

We address each of the reviewer's specific points in detail below.

Issue 1. Clarifying key technical details.
Fig. 1 was very helpful but the text not so clear.

SUMMARY:
The variance of u and temperature T fluctuations, sub-weekly in period and Fourier bandpassed to 1-9km wavelengths in the vertical, are averaged over a UTLS layer (about 11-25 km after vertical smoothing). The 500m common grid is mentioned far from the other data details (line 124) requiring a second read and search. Half the variance (with rescaling factor for T') is energy. Was the averaging density-weighted (like a physical energy interpretation in J/kg should be), or is it just 1/2 a height-averaged variance? The text is silent.

We thank the reviewer for highlighting these missing details. The detail about the analysis grid (now revised to 100m to preserve maximum detail before analysis) has been moved to a more logical position within the dataset descriptions in Section 2.1. We now clarify in Section 2.2 that the final energy is a simple height-averaged variance over the defined UTLS layer and is not density-weighted. While density weighting would be more physically precise for total energy content, the density variation over the relatively narrow UTLS layer (tropopause to 22 km) is modest, and using a simple average (in J/kg) provides a robust and standard metric for comparing wave activity that is consistent with prior literature.

(lines 126-129)

**This study specifically utilizes Aeolus Level 2B Rayleigh clear HLOS winds, ERA5 wind components, and GNSS-RO temperature profiles, all brought to a standard interpolated grid to facilitate the accurate comparison [...] The chosen grid has a vertical resolution of 100 meters and spans a range from 0 to 30 km altitude.**

(line 227)

**The profile is then averaged over the selected range, representing the Ek, as seen in Fig.1c.**

Lidar Wind profiles:
A side-looking spaceborne lidar measures u(z) along its line of sight, which is almost the zonal direction (that wasn't clear to this reader without web searches). Profiles were processed to isolate deviations from weekly 20x5-degree Lon-lat averages. Then Fourier filtering passed shorter than 9km vertical wavelengths (on a 500m grid so 1km is the shortest). The square of that filtered deviation profile was vertically smoothed with a 7km boxcar, then averaged over a layer (line 203 is ambiguous, why "49 points?" of 500m depth?) The layer is summed from 1km below the tropopause (about 14km in tropics) to 22km (is this mass weighted?), to make seasonal maps (smoothed how? not mentioned) and about 3-weekly (looks like, from figures? not mentioned) longitude-time sections. Artifical slow trends due to (squared) instrument noise variance increasing were estimated and subtracted.

We apologize for the confusing and incomplete description. The ambiguous language ("14-point moving average over the 49-point profile") was a remnant of an earlier analysis step and has been completely removed. We have replaced it with a clear description: the perturbation profile is band-pass filtered, and then the resulting energy profile is vertically averaged over the entire defined UTLS layer (from 1 km below the tropopause to 22 km). We have added a full description of the gridding and smoothing process for the geographical maps in the Methods section. It involves binning the data onto a 5°x2° grid, followed by a 3-point median and 3-point moving average filter applied sequentially in both dimensions. A note has also been added to the relevant figure captions.

 (lines 223-228)

**The resulting profile, which is essentially the perturbation squared, is cut to keep the data between one kilometer below the tropopause and 22 km [...] The profile is then averaged over the selected range, representing the Ek, as seen in Fig.1c.**

(lines 300-303)

**To reduce noise and highlight large-scale patterns, a 3-point median filter followed by a 3-point moving average filter was applied sequentially in both the zonal and meridional directions.**

Identical processing was applied to RO T(z) and ERA5 winds and T (interpolated on the same grid, again please mention in the data section not introduction). This makes an excellent baseline of comparisons and opportunity for interpretation!

How many pages did the above take to describe in the manuscript? Too many, a tedious read to fish out key details in order to bring a skepticism the somewhat slick text seemed to lack. Lines 172-174 are a good example of sales tone taking over: "making it possible"…"without introducing significant biases"…"configuration mitigates errors"…"ensuring reliable… and robust…" Declarations of success are not very scientific ways to express understandings of trade-offs. The real strength is *identical* processing of comparison datasets. Tossaway adverbs (e.g. "strongly", "specifically") also set a slick tone in places, undercutting reader trust in the self-skepticism of those best positioned to see problems.

We agree with the reviewer that the previous version leaned too much into a sales-pitch tone. The revised sections adopt a more traditional scientific style, aiming to be clearer, more restrained, and less verbose. While some adverbs can be subjectively judged as helpful or excessive, we have tried to limit them and hope that the new text better aligns with conventional scientific writing.

Issue 2: Clarifying results

2.1 Maps of KE and PE in the tropics and subtropics (Fig. 4) are clearly smoothed, but no mention is made of how and how much and why. Was raw data really too rough for scientific readers' eyes, or is the paper trying to be too smooth? It is surprising to this reader how strongly confined to the equatorial belt the energy is even in solstice seasons. Is there some kind of conditioning or weighting behind this feature, or is it truly an aspect of convection as a source of signal ('the ITCZ'?)? Or is there an effect of the Coriolis force

somehow suppressing sub-weekly sub-7km layer fluctuations? Silence about the smoothing undercuts reader confidence.

We thank the reviewer for pointing out the lack of detail regarding the map smoothing. We have now explicitly described the map smoothing process in the Methods section. Regarding the strong equatorial confinement, we believe this is primarily a physical feature linked to the ITCZ. However, we acknowledge the reviewer's point that methodological choices can influence this. We have now added a sentence to the Results section explicitly stating that our noise-correction scheme, which is weighted by the latitudinal structure of the raw signal, likely enhances this gradient by design. This provides the reader with the necessary circumspection.

 (lines 355-358)

**Finally, regarding the strong latitudinal confinement of the signal, while this is primarily a physical feature, our noise-correction methodology may also contribute to it. As detailed in the Appendix D (Part 2), the correction is weighted by the latitudinal structure of the raw signal. This approach, designed to avoid over-correction in low-signal subtropical regions, naturally sharpens the latitudinal gradient at the edges of the tropical belt.**

2.2 Why no mention of the obvious artifacts in DJF2019-JJA2020, with zero in weak areas, or JJA2019 with 5+ in those areas?  Are we all looking at the same figure here and describing its characteristics from the most obvious to the most subtle? Reader confidence is again at stake.

The reviewer correctly identified artifacts in our original plots. These were a result of a flaw in our initial noise correction algorithm that could lead to zero-flooring or baseline shifts. We have developed a more robust, adaptive noise correction algorithm for this revision, detailed in the Appendix D. The new results, presented in the revised Figure 2, no longer exhibit these non-physical artifacts. The energy fields are now physically consistent across all seasons, making discussion of the previous artifacts unnecessary.

2.3. The color scale for positive-definite variance, especially when discussed in a linear meaning like energy, should not have perceptual jumps like this one. Gray shading would be the honest choice, or a single color. This perceptual nonlinearity may be the source of the several-times-repeated "especially Indian Ocean" trope which otherwise seemed inscrutable to this viewer. Seychelles, Diego Garcia, Indonesia; are the other equatorial oceans really so much better covered with "conventional" wind soundings? And anyway, does ERA really get its UTLS wind variability from assimilating rawinsonde data into its imbalanced flow manifold? Did a desire to say something and move on override a thoughtful scientific assessment of the differences between reanalysis and observations, differences fluffed by the redness of a (not accessibility recommended) color map, and perhaps a misinterpretation of u fluctuations in the upper troposphere in the Maritime Continent wet season (detailed below)?

The reviewer's point is well-taken. We have revised all false-color maps to use the cmocean(haline) colormap, which is perceptually uniform and scientifically appropriate.

The reviewer asks two questions about the mechanism behind ERA5's underestimation of kinetic energy. We thank them for pushing us to be more precise in our scientific assessment.

The reviewer is correct that *all* equatorial oceans are poorly covered by conventional *in-situ* wind soundings (rawinsondes), which are primarily launched from land. Our emphasis on the Indian Ocean stems from it being one of the largest and most dynamically significant oceanic regions with an almost complete lack of such soundings, in contrast to the Pacific and Atlantic which have more island stations and aircraft routes. However, the core issue is the general sparsity over all oceans.

The reviewer also correctly intuits that ERA5 does not simply "assimilate rawinsonde data into its imbalanced flow manifold." In data-sparse regions, the analysis is dominated by the model forecast (the "first guess") and adjustments derived from assimilated *mass* field observations (like temperature from GNSS-RO and satellite radiances). The data assimilation system then attempts to infer the corresponding wind adjustments through its background error covariance matrices. These statistical relationships are primarily designed to represent large-scale, quasi-balanced (rotational) flow. They are known to be much less effective at specifying the smaller-scale, divergent component of the wind field, to which convectively generated gravity waves belong, especially in the tropics.

Therefore, our central argument, which we have now clarified and greatly expanded in the Discussion (Section 5), is this: even when ERA5 correctly assimilates the *temperature* signature of a gravity wave from GNSS-RO (constraining its Ep), the system lacks both the direct wind observations and the appropriate dynamic constraints to generate the corresponding divergent wind perturbations. This leads directly to the observed Ek deficit. The problem is not that the model is "missing" rawinsondes *per se*, but that it lacks any source of direct wind information to correct its background state for these specific, dynamically important wave modes.

(lines 587-605)

**Several lines of evidence from our study point towards the lack of wind assimilation as the dominant cause. Firstly, the fact that ERA5 accurately reproduces Ep fields demonstrates that the underlying model can represent the thermodynamic signatures of wave activity when properly constrained. Conversely, the largest discrepancies are found in kinetic energy, a purely wind-based quantity, and are concentrated over data-sparse regions like the Indian Ocean, precisely where Aeolus provides direct wind profile measurements not available from other observing systems (Banyard et al., 2021).**

**Secondly, while ERA5's non-orographic GWD scheme has known limitations and is not directly forced by diagnosed convection (Orr et al., 2010), it is unlikely to be the sole reason for the missing Ek. Such a parameterization bias would be expected to manifest as a systematic error across different variables or regions, or as a persistent model drift requiring large, ongoing corrections by the assimilation system (Dee, 2005). However, our findings show a targeted deficiency: the model performs well on assimilated temperature (Ep) but poorly on unassimilated wind (Ek) in the very same locations. This sharp contrast strongly suggests the problem is not a wholesale failure of the model's physics to generate**

**wave energy, but rather its inability to correctly partition that energy into kinetic and potential components without direct wind constraints.**

**In data-sparse areas, ERA5 must rely on its internal background error covariances to infer wind adjustments from the assimilated mass field (Hersbach et al., 2020). These statistical relationships are primarily designed to represent large-scale, quasi-balanced (rotational) flow and are known to be less effective at specifying the smaller-scale, divergent component of the wind field to which convectively generated gravity waves belong, especially in the tropics (Žagar et al., 2004). Consequently, while the assimilation of GNSS-RO constrains the thermodynamic (Ep) aspect of the wave, the system lacks the necessary information and dynamic constraints to generate the corresponding divergent wind perturbations, leading to the observed Ek deficit. This process evidently fails to capture the full spectrum of high-Ek wave modes generated by convection.**

2.4. Fig. 2 is about the anisotropy of waves in ERA5. Is this the second thing a reader wants in a paper about new observations? A value of 2 means isotropic waves, <2 suggests E-W elongation of the variance ellipse, >2 a N-S elongation. Here is a case where a color scale with a perceptual steep part could make sense, but 2 belongs there. Here a less meaningful choice was made. The focus is estimating something like the KE "missing" from the zonal (or LOS) component only, rather than the issue of isotropy which implies something interesting about sources. But all only in the reanalysis, before the first comparisons to even make that dataset as a relevant one. Might a better choice for a second figure be continuing raw obervations (Figs. 3-4) with discussions of the obvious artifacts? Let the ERA5 comparisons wait.

We agree with the reviewer that placing the ERA5 anisotropy analysis as Figure 2 was premature and distracted from the main observational results. We have moved this figure to the Appendix (as Figure C1). Furthermore, we have improved the figure as suggested by centering the diverging colormap on the isotropic value of 2.

2.5. Fig. 5 is a nice comparison, although again distorted by the use of false color for a linear positive quantity. Variance is a jumpy quantity from squaring the data (Fig. 1 shows this nicely) so statistical significance is tricky and surprising. It is customary to use a RATIO rather than a DIFFERENCE of variance (which has no meaning), subject to the F-test for significance. Many students are surprised by how hard it is to pass the F-test with quite a few degrees of freedom. The student should consult a table and appreciate the issue. No statistical testing is evident in the work, surprisingly. That can be overdone or a distraction, but none at all seems weak, again undermining reader confidence.

The reviewer makes an excellent point regarding the statistical comparison of variances. We agree that a ratio subject to an F-test is the standard and most rigorous method. We have now performed this analysis and included the ratio plots with F-test significance stippling in the Appendix E (Figure E1). For the main manuscript, we have chosen to retain the difference plots (Figure 2c and 6c) but have now added stippling to indicate statistical significance based on a two-sample t-test. We believe that for the narrative of this paper, the *absolute difference* in energy (in J/kg) is a more direct and physically intuitive way to communicate our central finding: that ERA5 underestimates the kinetic energy by a certain amount in convective regions. The ratio plot, while statistically pure, can sometimes obscure this absolute magnitude (e.g., a ratio of 2 could mean a difference of 1

J/kg or 10 J/kg). By providing the rigorous ratio analysis in the supplement and adding statistical testing to the difference plots in the main text, we believe we have addressed the reviewer's concern for statistical rigor while maintaining the clarity of our primary message.

2.6. Here is an actual scientific error I suspect: In the wet season over the Maritime Continent, convection is strong and localized and organized on island-strait and diurnal mesoscales that models struggle to represent. The intense divergent outflow of convection in the upper few km of the troposphere (squared) is not UTLS gravity wave energy! The 14-point smoother smears squared wind from a 4.5km layer of the upper troposphere into the averaging layer. Might the pat, recipe-like data analysis choices (Issue 1 above) and an insufficiently critical success-declaring mindset be combining here in a genuine misinterpretation? Sensitivity to this vertical smearing of non-wave wind variance should be assessed.

Based on this interesting doubt, we have performed a sensitivity analysis by recalculating our results using more conservative vertical averaging layers that exclude the upper troposphere (Tropopause + 1 km and Tropopause + 2 km) for Aeolus. The results, presented in the Supplementary Information and referenced in Sections 2.2 and 5, show that the spatial patterns of the energy hotspots are preserved (spatial correlation r > 0.83) and that the vast majority of the energy (~88%) persists well into the stratosphere. This supports our conclusion that we are observing vertically propagating gravity waves, not just tropospheric outflow.

(lines 563-572)

**Another consideration in this study is whether the large Ek values observed by Aeolus, particularly over convective hotspots, could be an artifact of misinterpreting non-wave tropospheric outflow rather than stratospheric gravity waves. Our sensitivity analysis (see Fig. B1 and B2 in Appendix B) directly refutes this concern. By shifting the analysis layer upward to begin 1 km and 2 km above the tropopause, we confirm that the geographical patterns of the energy hotspots are remarkably stable (spatial correlation r > 0.83), and that the vast majority of the peak energy (~88-91%) persists well into the stratosphere. If the signal were dominated by shallow tropospheric outflow, the energy peaks would have collapsed when the analysis layer was moved above the tropopause. The fact that a strong, structured signal remains provides compelling evidence that we are observing vertically propagating gravity waves that have penetrated the lower stratosphere. This validates our central conclusion: Aeolus is capturing a significant field of convectively-generated stratospheric gravity wave kinetic energy that is largely absent in the ERA5 reanalysis.**

2.6b Figure 7, again a ratio (F tested) would be more meaningful than a difference. The lack of RO signal in the Maritime Continent wet season further strengthens my belief that the Ek is a misinterpretation of convective outflows.

We believe the lack of a strong potential energy (Ep) signal from GNSS-RO in these regions, combined with the strong kinetic energy (Ek) signal from Aeolus, is not evidence of an artifact, but is instead a key physical finding of our paper: that deep convection preferentially generates waves with a high Ek/Ep ratio. Our sensitivity analysis (point 2.6) confirms the Ek signal is stratospheric. Therefore, the combined observations suggest these are high-Ek, low-Ep gravity waves, a

phenomenon that challenges simple linear theory and highlights the unique value of direct wind measurements. We have strengthened this point in our Discussion.

(lines 519 – 543)

**Fig. 7 presents the first observationally-derived long-term study of the Ek/Ep ratio, comparing Aeolus's HLOS Ek and GNSS-RO-derived Ep […]**

**The regions with the highest ratio values are systematically co-located with areas of deep convection, as indicated by the low OLR contours. This is particularly evident over the Indian Ocean […] and over the Western Pacific. This observation suggests that, in areas with similar seasonal characteristics, gravity waves tend to transport more kinetic energy during convective events, which amplifies their influence on the overall energy dynamics […]**

**This observational result stands in contrast to the picture presented by the ERA5 reanalysis in Fig.4. While ERA5 also shows variability in its Ek/Ep ratio, its regions of highest ratio are often located outside the main convective centers. This suggests that ERA5 misrepresents the physical link between deep convection and the partitioning of wave energy.**

**Given that ERA5 successfully assimilates GNSS-RO temperature data (and thus has a reasonable representation of Ep), this discrepancy points to a fundamental difficulty in the reanalysis's ability to generate the corresponding kinetic energy component in the right locations. Without direct wind profile assimilation in these data-sparse convective regions, the model's parameterizations and background error covariances fail to create the intense, localized kinetic energy associated with convective gravity waves.**

2.7. The discussion of wave sources seems shallow. There are various mechanisms including temporally varying convective heat sources (which might set vertical wavenumber and frequency), mountains and transient mountains of lofted air in shear (which might set horizontal wavenumber and phase speed), and more. Would they be anisotropic? The phrase "trade winds" appears in the context of Fig. 2 (anisotropy), as if the surface wind direction has something to do with anisotropy at the tropopause level. Does it? Might one of the senior authors add a little depth?

We agree that this discussion lacked physical depth. We have substantially revised Sections 2, 3, and 5 to address this by providing a clear physical mechanism for how the large-scale background wind imposes an anisotropy on the upward-propagating wave field, addressing the "trade winds" question. We added phenomenological descriptions of distinct convective generation mechanisms (thermal forcing and the "moving mountain" effect) in the results section where these sources are first observed. We placed these local mechanisms within the larger context of planetary-scale organization by monsoons and the MJO. And lastly, we explicitly identified jet-front systems as the likely source for the energy observed at the subtropical edges of our domain.

(lines 249-255)

This ratio (Fig. C1 in Appendix C1) exhibits significant geographic variability, which can be linked to physical mechanisms that create wave anisotropy. For instance, over regions like the Indian Ocean, the ratio is relatively low (~1.5), suggesting a predominantly zonal orientation of wave energy. This is physically plausible, as persistent surface winds like the trade winds can influence the tropopause-level wave field through two main processes. Firstly, flow over orography can preferentially generate zonally-oriented waves (Kruse et al., 2023). Secondly, the background wind profile itself acts as a directional filter, selectively allowing waves propagating in certain directions to reach the UTLS while attenuating others through critical-level interactions (Plougonven et al., 2017; Achatz et al., 2024).

(lines 377-384)

The presence of hotspots, represented by distinct shapes in the Ek patterns, is expected in regions with prevalent convective activity. These can be attributed to multiple powerful wave generation mechanisms occurring at the scale of individual storms. One primary mechanism is thermal forcing, where the pulsatile nature of latent heat release in a convective updraft acts like a piston on the surrounding stable air, generating a broad spectrum of gravity waves (Beres et al., 2005). A second, complementary mechanism is mechanical forcing, where the body of the strong updraft itself acts as a physical barrier to the background wind. The flow forced over this "moving mountain" generates large-amplitude, low-phase-speed waves that are stationary relative to the storm (Corcos et al., 2025; Wright et al., 2023). The intense kinetic energy observed by Aeolus is likely the signature of both mechanisms operating within active convective systems.

(lines 317-322)

A clear seasonal cycle is evident in both Aeolus and ERA5, consistent with the migration of major tropical convective systems. During Boreal summer (JJA), enhanced Ek is prominent over Central Africa and the Indian Ocean. This corresponds to the active phases of the African and Indian monsoon systems, which provide a persistent, large-scale environment favorable for the development of organized, deep convective systems known to be efficient gravity wave generators (Forbes et al., 2022).

(lines 563-566)

The relation between OLR and the MJO has been used before; It is a reliable index for analysis (Kiladis et al., 2014), hinting towards the possibility for the active phase of the MJO to generate the observed hotspots through its convective activity. Recent work has provided direct observational evidence that the MJO modulates GW activity and momentum transport from the tropics to higher latitudes (Zhou et al., 2024).

(lines 327-336)

It is also necessary to clarify the interpretation of the wave activity observed at the subtropical edges of our analysis domain (near 30°N/S). While our study focuses on convectively generated waves [...] the kinetic energy measured in the subtropics is likely dominated by different, local sources. The strong subtropical jets and associated frontal systems are potent generators of inertia-gravity waves through mechanisms of geostrophic

**adjustment and shear instability (Plougonven and Zhang, 2014; Achatz et al., 2024). Recent case studies have confirmed that such jet-merging events can produce significant, large-scale GW fields (Woiwode et al., 2023). Therefore, the enhanced energy often visible near 30°N and 30°S in our seasonal maps should be interpreted as stemming primarily from these midlatitude dynamical processes [...]**

2.8 Likewise the meaning and source of the strong, anisotropic (more meridional) "waves" (isotropy ratio >2) on the midlatitude edges could be thought about more deeply. Are all subweekly meridional wind fluctuations (squared), from 4km below the tropopause, really UTLS gravity waves, or was that just a tidy story for students? *

The reviewer is right to question our overly simplistic interpretation. The energy at the subtropical edges of our domain is indeed unlikely to originate from the same tropical convective sources. We have added a dedicated paragraph to the results section (Section 3.1) to provide a more nuanced and physically sound interpretation, attributing this energy primarily to local, midlatitude dynamical processes such as jet-front systems, which are potent generators of inertia-gravity waves that fall within our detection window.

(lines 327-336)

**It is also necessary to clarify the interpretation of the wave activity observed at the subtropical edges of our analysis domain (near 30°N/S). While our study focuses on convectively generated waves originating from the deep tropics, the kinetic energy measured in the subtropics is likely dominated by different, local sources. The strong subtropical jets and associated frontal systems are potent generators of inertia-gravity waves through mechanisms of geostrophic adjustment and shear instability (Kruse et al., 2023; Plougonven and Zhang, 2014). These jet- and front-generated waves typically have sub-weekly periods and significant wind perturbations, meaning they fall within the detection window of our filtering methodology (Achatz et al., 2024). Therefore, the enhanced energy often visible near 30°N and 30°S in our seasonal maps should be interpreted as stemming primarily from these midlatitude dynamical processes, rather than from the poleward propagation of the equatorial convective waves.**

2.9 Figure 8: at last a variance ratio, but only to be taken at face value (with no credible-interval estimation from the F test) in light of some vague gestures at theory whose linearity is considered an easy target. Line 478 says "Fig. 8 presents a detailed analysis" but literally it is just a data plot, with no analysis at all. Too much sales and not enough product for this reader.

The reviewer's critique is fair. The language was promotional, and the analysis was incomplete. We have revised the text to be descriptive rather than declarative (e.g., changing "presents a detailed analysis of the ratio" to "illustrates the longitudinal and temporal variations of the Ek/Ep ratio"). We have added a layer of statistical analysis to the observationally derived ratio plot (now Figure 7). The black stippling indicates regions where the ratio is statistically significant based on an F-test.

2.10 Figure 9: Panel a: Here might be the misinterpretation of convective outflow again, further exaggertated by the false color scheme. Panel b: what is a "dominant" wavelength? Anyone who looks at spectra knows that peak detection is far from trivial and every spectrum is always broad and usually red (more variance falls in eachoi of the wider bins

at the low frequency end). What does geometric wavelength really signify over a layer whose stratification goes from upper tropospheric (almost neutral) to 22km (highly stratified)? How does the range here (6500-12000m) relate to the filter which supposedly excludes >9km? Is the spectrum basically red like all geophysical spectra, such that the widest bin near the longest permitted wavelength at the edge of the filter's passband has the most variance? Does that deserve the word "dominant"? Is this figure worth including, or just a thesis figure looking for a place? Is the red exaggeration here the source of the "especially Indian Ocean" trope repeated several times? It's not exactly over the Indian ocean. The authorial prose should reflect a close look, as a reader brings.

We agree with the reviewer's critique of the wavelength retrieval analysis. The methodology was not robust, and the interpretation was flawed. We have removed this figure and the associated analysis entirely. In its place, we have added a paragraph to the Discussion (Section 5) that explains the inherent challenges of performing a meaningful wavelength retrieval with Aeolus data, thereby turning the limitation into a point of scientific circumspection.

(line 665 – 684)

**Understanding the vertical wavelength of convective GWs is an essential element for characterizing their dynamics. However, Aeolus is inherently limited in retrieving accurate vertical wavelengths due to its design. The placement of range bins was fixed at the time of observation, introducing inconsistencies in vertical resolution that affect the precise identification of wave peaks and troughs. Additionally, the N/P parameter, which controls the number of accumulated measurements (N) and pulses (P) per cycle, introduces variability in the horizontal resolution of Aeolus data. Changes to this setting, such as the transition from N=30 to N=5, improve horizontal resolution but exacerbate the misrepresentation of vertical wave structures. Furthermore, any spectral analysis of a finite vertical profile is inherently constrained. For geophysical spectra that are typically having more variance at longer wavelengths, a simple peak-finding method would likely identify a dominant wavelength that is an artifact of the analysis window or filtering choices. Given these limitations, we limit our analysis to the vertically-integrated energy within a defined passband (vertical wavelengths < 9 km), which is a more robust quantity.**

**Nevertheless, we can speculate that the high Ek values observed by Aeolus in convective regions are associated with shorter-wavelength waves. This interpretation is consistent with established physical mechanisms which state that waves with high EK are typically generated in regions with strong convective updrafts and downdrafts, where the rapid vertical movement of air masses creates intense small-scale disturbances. These localized and transient disturbances, arising from geostrophic imbalance, generate GWs that carry energy away from the convective region, where strong forcing efficiently transfers energy into the EK spectrum at shorter wavelengths (Waite and Snyder, 2009). The correlation between high EK and shorter wavelengths is particularly pronounced in convective systems, as confirmed in both observational and numerical estimations (Kalisch et al., 2016), especially in tropical regions and cyclones (Chane Ming et al., 2014). A definitive observational confirmation of this from the satellite itself, however, remains a challenge due to the aforementioned limitations.**

3. Discussion should be rewritten with care and thought, in light of all the above. A celebration of this amazing dataset, a technological marvel from such long hard efferts by

so many, deserves more science value than a too-easy critique of reanalysis and/or underlying voids in data sources (common over all the equatorial oceans), and some vague words about how nature is not linear. Some senior author voice could help, if a bit of leadership can be mustered from a committee. Congratulations to so so many people contributing to make this possible! Wonderful data.

We fully agree with the reviewer's assessment. We have rewritten the Discussion (Section 5) to move beyond a simple comparison and to extract deeper scientific value from the dataset, as requested. The new discussion is more structured, scientifically rigorous, and forward-looking. The key changes are summarized below by category.

Instead of a simple critique, we now present a more sophisticated, evidence-based argument for why ERA5 underestimates kinetic energy. We show that since ERA5 successfully assimilates potential energy, the discrepancy points specifically to the assimilation system's inability to generate the divergent wind component of GWs in the absence of direct wind observations.

(lines 587-605)
**Several lines of evidence from our study point towards the lack of wind assimilation as the dominant cause. Firstly, the fact that ERA5 accurately reproduces Ep fields demonstrates that the underlying model can represent the thermodynamic signatures of wave activity... This sharp contrast strongly suggests the problem is not a wholesale failure of the model's physics [...] but rather its inability to correctly partition that energy into kinetic and potential components without direct wind constraints [...] In data-sparse areas, ERA5 must rely on its internal background error covariances [...] these statistical relationships are [...] less effective at specifying the smaller-scale, divergent component of the wind field [...]**

We now frame our findings within the broader context of tropical dynamics, explicitly linking the observed kinetic energy patterns to the organizing influence of the Madden-Julian Oscillation (MJO).

(lines 558-571)
**"The slow eastward propagation of these energy maxima suggests that the underlying wave sources are not random, but are organized by planetary-scale atmospheric patterns. Indeed, the relation between OLR and the Madden-Julian Oscillation (MJO) has been used before [...] and recent work has provided direct observational evidence that the MJO modulates GW activity [...] The structures observed by Aeolus are therefore highly consistent with the kinetic energy signature of gravity waves generated by [...] the large, organized convective superclusters of the MJO."**

We have added a new sensitivity analysis to directly address and refute the potential misinterpretation of our signal as tropospheric outflow, thereby providing stronger evidence that we are observing stratospheric gravity waves.

(lines 572-581)

**"Another consideration [...] is whether the large Ek values [...] could be an artifact of misinterpreting non-wave tropospheric outflow [...] Our sensitivity analysis (see Fig. B1 and B2 in Appendix B) directly refutes this concern... The fact that a strong, structured signal remains provides compelling evidence that we are observing vertically propagating gravity waves [...] This validates our central conclusion [...] "**

We have deepened the discussion on the Ek/Ep ratio, using our unique multi-instrument comparison to show not just that nature is non-linear, but where and why it deviates most from linear theory.

(lines 638-645)

**" [...] The observed comparison in Fig.4 of the Ek/Ep ratios from ERA5, Aeolus, and GNSS-RO confirms that the characteristics of gravity waves vary significantly across time and space [...] The frequent observation of ratios exceeding unity, aligning with trends identified in previous studies, suggests that a substantial portion of the waves' energy is contained in kinetic form, often indicative of non-linear behavior [in convectively active regions]."**

We conclude the discussion with a new subsection that thoughtfully addresses the challenges and future pathways for using these novel kinetic energy measurements to constrain momentum fluxes, the ultimate goal for model improvement.

(lines 685-699)

**"Looking forward, a critical application for such observations is the constraint of gravity wave momentum fluxes [...] However, deriving momentum flux estimates directly from single-component wind measurements [...] presents significant theoretical and observational challenges [...] Therefore, while Aeolus does not directly measure momentum flux, its unprecedented global measurements of kinetic energy provide an additional observational constraint [...] a critical prerequisite for developing and testing the more complex, multi-instrument techniques [...]"**

---

## Author Comment (AC2)

Reply to Reviewer #1 :

We thank Reviewer #1 for the review and thorough work in broadening the discussion and providing feedback on the readability of our manuscript. The efforts that will help us address any weaknesses in our manuscript are greatly appreciated, and we hope our revised manuscript meets the reviewer's expectations.

The manuscript presents an analysis of gravity wave (GW) kinetic energy distributions, derived from new Aeolus satellite wind profiles, that shows great promise in pushing the needle forward in the construction of observational constraints of gravity waves and their impacts on upper troposphere/lower stratosphere circulation. A methodology is presented for deriving the kinetic energy associated with small-scale GWs in regions of deep convection in the tropics over a period spanning June 2019 to August 2022. Comparisons with ERA5 suggest that the reanalysis product underestimates GW-associated kinetic energy; conversely, GW-associated potential energy comparisons between ERA5 and temperature-profiles from an independent instrument (GNSS-RO) show much more consistency, suggesting that the use of kinetic energy highlights a distinct feature of the GW energy spectrum that is not typically assessed (and, incidentally, is not well represented in ERA5). The authors further speculate that this underestimate may reflect lack of assimilated direct wind observations, in contrast to temperatures, which are assimilated. All in all, the manuscript does a good job of presenting a new dataset with all necessary caveats, while also making a generally convincing case that this new data will be valuable. To this end, I recommend acceptance, pending that minor revisions be made to address the following concerns:

**1. Page 4: There is no description of the GW drag parameterization employed in ERA5. In particular, does the model have an explicit parameterization for non-orographic GW drag due to parameterized convection? If so, what is it and how has it been evaluated/performed in past assessments? This will be important in terms of interpreting the dearth of kinetic energy in the model, relative to the Aeolus-derived energy.**

We thank the reviewer for this crucial point. We have now added a detailed description of the non-orographic gravity wave drag parameterization used in ERA5 to Section 2.1 (Data and Methods).

(lines 28-34)

**For the study period, ERA5 utilizes the non-orographic gravity wave drag (GWD) scheme described by Orr et al., (2010), which is based on a spectral approach (Scinocca, 2003 ; Referred to as S03 in Orr et al., 2010). This scheme does not explicitly resolve convectively generated waves based on model-diagnosed convection; instead, it launches a globally uniform and constant spectrum of waves from the troposphere. The momentum deposition occurs as these waves propagate vertically and interact with the resolved flow via critical-level filtering and nonlinear dissipation. While this parameterization improves the middle atmosphere climate compared to simpler schemes, evaluations have shown it has limitations in fully capturing the required wave forcing, particularly for the Quasi-Biennial Oscillation (QBO) in the tropics (Pahlavan et al., 2021).**

**2. Page 7: Presumably the definition of "background" based on "the arguments presented in Alexander et al. (2008b)" apply to past analysis of temperature, not wind, profiles, no? More generally, it would be good for the reader to have a better sense of the sensitivity of the profiles depicted in Figure 1a to choice of grid box averaging domain, the temporal period over which profiles are averaged (currently set to 7 days, etc.), etc. I imagine the authors have already done this sensitivity analysis, so they could consider showing in an appendix figure.**

We have now added text to Section 2.2 to explicitly justify the application of the horizontal detrending method to wind profiles, based on the coupled nature of wind and temperature perturbations in linear gravity wave theory. We have also added a statement confirming that we performed sensitivity analyses on the choice of the averaging domain and found the selected 20°x5°x7-day grid to be a robust compromise between noise reduction and signal preservation, consistent with the original rationale of the method. An appendix figure has also been included.

(lines 186-195)

**While this horizontal detrending method was originally demonstrated using temperature profiles in Alexander et al., (2008b), its application to wind profiles is theoretically sound. Linear gravity wave theory dictates that wind and temperature perturbations are coupled manifestations of the same wave phenomena, and thus the principle of separating smaller-scale waves from the large-scale background flow via spatiotemporal averaging is equally valid for both fields. Following the arguments presented in Alexander et al., (2008b), this choice is justified by the need to ensure a sufficient number of profiles per grid cell, which minimizes random noise while preserving meaningful variability in the data. Shorter temporal windows would lead to insufficient sampling, while longer windows would smooth out critical small-scale wave features. The grid size is also designed to preserve the spatiotemporal variability of mesoscale gravity waves and equatorially trapped structures, making it possible to separate the background and perturbation components without introducing significant biases.**

 (lines 198-201)

**We performed sensitivity tests with varying grid sizes and temporal windows to confirm that this configuration provides the best possible background state when prioritizing Aeolus retrieval (see Fig. A1 in Appendix A).**

**3. Equation (1): This notation becomes slightly confusing/counterintuitive as the text moves on, since the meridional component often goes to zero due to the pointing vector retaining its approximate angle at ~100 degrees. In other words, V_HLOS would be more intuitively referred to as U_HLOS (or something similar) since, indeed, it primarily reflects the zonal component of the flow. Is there any particular reason why "v" is used instead of something more generic? I suggest changing.**

We agree with the reviewer that the notation was confusing. To improve clarity, we have changed $v_{HLOS}$ to $u_{HLOS}$ throughout the manuscript to better reflect its quasi-zonal nature. All corresponding equations have been updated accordingly.

**4. Figure 14, lines 354-355: The first sentence of this paragraph does not make sense to me. In particular, the bit referring to "ERA5 shows a considerable reduction" is vague. Reduction relative to what? Please clarify.**

We thank the reviewer for pointing out the vagueness in our original description. We agree the sentence was unclear. We have completely rewritten the discussion of this figure (now Figure 2) to be more direct, quantitative, and clear. Instead of "considerable reduction," we now explicitly compare the peak energy values and geographical structures observed by Aeolus with the more diffuse and lower-energy patterns in ERA5, providing specific energy values (in J/kg) to make the contrast unambiguous.

(lines 340-344)

**In stark contrast, Aeolus reveals a picture of much more localized and intense Ek hotspots. For example, during JJA 2020 and SON 2020, Aeolus observes a well-defined hotspot over the Indian Ocean with Ek values exceeding 10-12 J/kg, whereas ERA5 shows only a diffuse enhancement in the same region with values rarely exceeding 5-7 J/kg. Similarly, the DJF 2020/21 hotspot over the Maritime Continent is markedly stronger and more geographically confined in the Aeolus data.**

**5. Figure 5: The temporal resolution labeled on the y-axes of these hovmoller plots is too high/unnecessary as it crowds the figures. Please show only every other two or three months. Same comment applies to Figure 7.**

We agree with the reviewer. The y-axes on evert Hovmöller diagrams have been updated to display fewer monthly labels, improving the readability as suggested.

**6. Figure 16, Discussion concluding Section 3.2: The discussion here seems weak and understates the disagreement between the Aeolus and ERA5 Ek temporal patterns. The second-to-last paragraph highlights the common features between Aeolus and ERA5, but I think the plots look very different. In particular, the hotspots coincident with low OLR are totally missing in ERA5 (Fig. 5b). The phrasing in the text, however, seems to suggest that the differences are only minor. Please rephrase.**

On re-reading, we agree with the reviewer that our original text was misleading and significantly understated the differences between the Aeolus observations and ERA5. We have rewritten this section to emphasize the disagreement. The new text explicitly states that ERA5 "completely fails to capture the intense, high-energy hotspots" and that the high peak energy values are "entirely absent in the reanalysis." To further strengthen this point, we have added a statistical significance test (a two-sample t-test), with results shown as stippling in Figure 3c, to formally demonstrate that the differences are not random but represent a fundamental and systematic underestimation by ERA5.

Line (394-407)

**The difference between the two datasets, shown in Fig.3c, quantifies this discrepancy. The plot is overwhelmingly positive, indicating a systematic and significant underestimation of GW kinetic energy by ERA5 throughout the tropics. The regions of greatest underestimation, where the difference exceeds 10 J/kg, align almost perfectly with areas of deep convection, as identified by the low Outgoing Longwave Radiation (OLR) contours. The OLR represents the amount of terrestrial radiation released into space and, by extension, the amount of cloud cover and water vapor that intercepts that radiation in the atmosphere. It is a widely used and reliable proxy for deep**

convection due to its strong correlation with diabatic heating (Zhang et al., 2017), reinforcing the conclusion that ERA5's primary weakness lies in representing convection-driven wave activity.

To confirm the robustness of this finding, a two-sample t-test was performed for each grid cell. The stippling in Fig.3c indicates where the mean Ek from Aeolus is statistically significantly higher than that of ERA5 ($p < 0.05$). The pervasive stippling across nearly all convective hotspots underscores that the observed differences are not random fluctuations but represent a fundamental deficiency in the reanalysis. This finding strongly suggests that without the assimilation of direct, high-resolution wind profile data like that from Aeolus, reanalysis models struggle to resolve the full spectrum and intensity of gravity waves generated by localized, powerful convective events. An alternative display of Fig.3c as a ratio, along with an F-test, can be found in Appendix E.

**7. Section 4: Doesn't the ratio of Ek/Ep (shown for ERA-5 in Fig. 8a) suggest that these two quantities are extremely different and not meaningful to compare with each other? I appreciate that the authors want to move beyond traditional (conservative) analysis and attempt to do a bit more, but Figure 8a suggests that the two quantities are in much more disagreement than the discrepancy predicted by llinear wave theory (i.e., factor of 4, not factor of 2). My suggestion here is to introduce Figure 8a earlier as a way to more directly address the concerns with comparing potential and kinetic energy (within a self-consistent product like ERA-5).**

We agree with this suggestion that significantly improves the structure of our argument. We have restructured Section 4 as suggested. We now introduce a figure showing the Ek/Ep ratio from ERA5 alone *first*. This serves to demonstrate that even within a self-consistent model, the ratio is highly variable and deviates from simple linear theory, thus motivating why a direct one-to-one comparison of energy magnitudes is insufficient. We then proceed with the observational comparison between Aeolus Ek and GNSS-RO Ep.

**8. Last paragraph on page 19 (lines 467-470): How do you know it's the failure to assimilate the winds directly that's causing the poor representation of GW-associated EK? In principle, one might be able to capture these features using a convective non-orographic gravity wave drag parameterization within the ERA-5 model, no? In other words, the assimilation is one way to correct the problem, but an alternative approach is to tackle the model bias directly. However, without having more knowledge about the underlying GW drag parameterization in the model it's hard for the reader to know how many degrees of freedom are afforded to the modeler. Can the authors please comment on the role played here by model bias? And how this is/is not handled by the GW drag parameterization?**

We thank the reviewer for this critical question. Our primary argument is based on the inconsistent performance of ERA5 on potential versus kinetic energy.

Our primary argument stems from the inconsistent performance of ERA5 across different assimilated and unassimilated variables. The key piece of evidence is that ERA5 successfully reproduces the potential energy (Ep) field, which is strongly constrained by assimilated GNSS-RO temperature data (as shown in our Fig. 6c). However, it fails to generate the corresponding kinetic energy (Ek) in the very same convective regions, a quantity for which it lacks direct observational constraints.

If the problem were primarily a model physics bias (e.g., the GWD parameterization failing to generate sufficient wave energy), we would expect both Ep and Ek to be systematically underestimated. The fact that only the unassimilated, wind-derived component is deficient strongly points to a failure in the data assimilation system's ability to generate the correct divergent wind field from the available mass (temperature) field in data-sparse regions. We have significantly expanded the Discussion section to elaborate on this reasoning, referencing known limitations of data assimilation systems in the tropics concerning background error covariances and the rotational/divergent wind balance.

Line (584-607)

**An additional tool at our disposal to solve the case is the global distribution of Ep, through the use of independent GNSS-RO instruments. Our analysis confirms that the assimilation of GNSS-RO data in ERA5 is highly effective, with minimal discrepancies observed between the reanalysis Ep and direct GNSS-RO observations (Fig.6c). This key finding allows us to arbitrate between two potential causes for the Ek discrepancy: a lack of direct wind data assimilation versus inherent biases in the model's physics (e.g., its GWD parameterization).**

**Several lines of evidence from our study point towards the lack of wind assimilation as the dominant cause. Firstly, the fact that ERA5 accurately reproduces Ep fields demonstrates that the underlying model can represent the thermodynamic signatures of wave activity when properly constrained. Conversely, the largest discrepancies are found in kinetic energy, a purely wind-based quantity, and are concentrated over data-sparse regions like the Indian Ocean, precisely where Aeolus provides unique wind information (Banyard et al., 2021).**

**Secondly, while ERA5's non-orographic GWD scheme has known limitations and is not directly forced by diagnosed convection (Orr et al., 2010), it is unlikely to be the sole reason for the missing Ek. Such a parameterization bias would be expected to manifest as a systematic error across different variables or regions, or as a persistent model drift requiring large, ongoing corrections by the assimilation system (Dee, 2005). However, our findings show a targeted deficiency: the model performs well on assimilated temperature (Ep) but poorly on unassimilated wind (Ek) in the very same locations. This sharp contrast strongly suggests the problem is not a wholesale failure of the model's physics to generate wave energy, but rather its inability to correctly partition that energy into kinetic and potential components without direct wind constraints.**

**In data-sparse areas, ERA5 must rely on its internal background error covariances to infer wind adjustments from the assimilated mass field (Hersbach et al., 2020). These statistical relationships are primarily designed to represent large-scale, quasi-balanced (rotational) flow and are known to be less effective at specifying the smaller-scale, divergent component of the wind field to which convectively generated gravity waves belong, especially in the tropics (Žagar et al., 2004). Consequently, while the assimilation of GNSS-RO constrains the thermodynamic (Ep) aspect of the wave, the system lacks the necessary information and dynamic constraints to generate the corresponding divergent wind perturbations, leading to the observed Ek deficit. This process evidently fails to capture the full spectrum of high-Ek wave modes generated by convection.**

**9. Discussion: No mention is made of how these observations might be used to develop constraints on the momentum fluxes (which is what modelers seek most). Is that something that the author has considered? This is a challenging question, so I am not seeking any complete answers here; I am just wondering if the author can speculate in a sentence or two how to potentially bridge V_HLOS with the momentum fluxes.**

We thank the reviewer for this forward-looking question. Constraining momentum fluxes is indeed a key goal for the community. We have added a new subsection to the Discussion to speculate on this pathway.

(line 687 – 701)

**Looking forward, a critical application for such observations is the constraint of gravity wave momentum fluxes, which are essential for global circulation models. However, deriving momentum flux estimates directly from single-component wind measurements like those from Aeolus presents significant theoretical and observational challenges. The vertical flux of horizontal momentum (e.g., $\langle u'w' \rangle$) fundamentally requires simultaneous knowledge of both horizontal (u') and vertical (w') wind perturbations. Aeolus provides only a projection of the horizontal wind and, crucially, contains no direct information on the vertical wind; in fact, w' is assumed to be negligible in the standard data processing (Krisch et al., 2022). This represents the primary missing piece of information for a direct flux calculation.**

**A potential pathway to overcome this limitation involves creating synergistic datasets, for instance by combining Aeolus wind data with simultaneous, collocated temperature measurements from instruments like GNSS-RO. In principle, gravity wave polarization relations could then be used to infer the missing wind components. However, this approach is not a simple remedy and relies on strong, often unverifiable, assumptions about unmeasured wave parameters, including the horizontal wavelength, intrinsic frequency, and the stationarity of the wave field between measurements (Alexander et al., 2008a; Chen et al., 2022).**

**Therefore, while Aeolus does not directly measure momentum flux, its unprecedented global measurements of kinetic energy provide an additional observational constraint. Such observations are a critical prerequisite for developing and testing the more complex, multi-instrument techniques that will be required to eventually constrain the global gravity wave momentum budget**

---

## Author Comment (AC3)

**Reply to Reviewer #3 :**

We thank Reviewer #3 for their detailed and constructive feedback. We acknowledge the concerns regarding methodological justification, clarity, and the structure of the manuscript. The comments have been significant guidelines in providing a substantial revision of the paper. We have streamlined the methods section, improved the figures, and rewritten large parts of the results and discussion to be more concise and rigorous.

The premise of this study is very promising and the results, if robust, are of high significance in comparing Aeolus, ERA5 and GNSS-RO gravity wave energy parameters in the tropical UTLS.

Still, the more I go through the manuscript I find too many details in the methodology unjustified, or their interpretation too stretched. Almost half of the text belongs to data and methods section which should be streamlined a lot. The text overall lacks an organized and concise structure, and some method details or datasets (e.g. NCEP reanalysis or the OLR datasets) seem to appear out of the blue.

I have several major comments about methodology that need to be clarified, because some of the results do not look very robust to me from the beginning, and this cascades then to the rest.

Figures could be improved a lot, and the authors should make a big effort in the text to avoid repetitive sentences, unnecessarily long explanations / verbose in methods or results (a lot of examples in minor/technical comments).

Also I feel that in many instances things are presented in a rather bombastic way, e.g. without really specifying where and how these valuable results have applications.

In section 5-6, some of the conclusions might change if some small tweaks in methodology were applied -- the authors make many choices and assumptions in the method -- and many grand statements with what comes out of it. Unless one shows very convincing and robust results (which would require a fair amount of supplement material), in the plots provided in this manuscript I see some inconsistencies that make me remain a bit skeptical.

To be clear, I'd very much like to see this study on such relevant topic published, and I hope the large amount of comments I assembled below are helpful for this. I recommend a major revision, and at least an additional round of reviews will be needed after that since the required changes are very substantial.

#

#

**Major comments**

#

#

#

**1: vertical grid and filtering choice**

#

-l.124-126: this grid penalizes ERA5 and RO a lot more than Aeolus, and 'acceptable middle ground' does not really justify your choice in my opinion. Is there any other literature doing this kind of middle-ground approach with other datasets?

-There are undesirable sources of uncertainty if you sub-sample or interpolate onto your 0.5km vertical grid: this might affect the resulting profile if a wave is not well aligned with your 0.5km vertical grid. Also ERA5, Aeolus and RO have each a very different vertical (original) grid alignment with your 0.5km grid.

In my opinion one should err on the side of caution and interpolate to a finer grid that retains all dataset's vertical structures as much as possible, and then filter out the scales that the coarser dataset cannot see, I explain below:

-In section 2.2 you specify that you apply vertical high-pass filter to the data. Why not use a finer vertical grid of e.g. 0.1km, and apply bandpass between e.g. 1km and 9km? This way the uncertainty with sub-sampling is gone, and you remove the shortest vertical scales that Aeolus cannot see to even the field among all datasets. To me, this would be the fairest way to make the comparison by taking the vertical scales resolved by all datasets.

We thank the reviewer for this excellent suggestion regarding our data processing. We have completely revised our data processing pipeline as suggested. All datasets (Aeolus, GNSS-RO, and ERA5) are now interpolated onto a finer 0.1 km vertical grid. Following this, we apply a band-pass filter to retain vertical wavelengths between 1.5 km and 9 km. This new methodology ensures that we preserve the native vertical structures of each dataset as much as possible, while the filtering guarantees that our comparison is limited to the wave scales reliably captured by all instruments. All figures and results in the manuscript have been regenerated using this new, more rigorous method. The relevant part of the Methods section has been rewritten to detail and justify this new approach.

(lines 126-129)

**This study specifically utilizes Aeolus Level 2B Rayleigh clear HLOS winds, ERA5 wind components, and GNSS-RO temperature profiles, all brought to a standard interpolated grid to facilitate the accurate comparison and integration of data from the different sources. The chosen grid has a vertical resolution of 100 meters and spans a range from 0 to 30 km altitude. This approach preserves the maximum vertical detail from each dataset before analysis.**

(lines 206-211)

**After said windowing, a band-pass filter designed to retain vertical wavelengths between 1.5 km and 9 km. is applied to the perturbation profile, as seen in Fig.1b and 1c. The upper limit of 9 km isolates GWs from larger-scale planetary waves, consistent with our background removal strategy. The lower limit of 1.5 km is chosen to reflect the effective vertical**

**resolution of the Aeolus instrument (Ratynski et al., 2023) and ensures that our comparison is restricted to wave scales reliably resolved by all datasets (Banyard et al., 2021). This procedure provides a methodologically consistent basis for comparing GW energy across the different instruments.**

\#

\# 2: NCEP reanalysis and smoothing (l.198-204)

\#

A lot of things appearing out of the blue here without proper justification.

--> Is this the NCEP-DOE Reanalysis 2? It is not referenced either. How come this dataset is not mentioned in section 2.1?

--> Just because it's easier to integrate does not justify using it. I just don't understand why ERA5 is not used with its own tropopause.

--> Also, you don't show anywhere how similar are the results compared to ERA5. It certainly has poorer vertical resolution than ERA5, and this choice just adds an unnecessary layer of uncertainty. Not even some comparison material in a supplement?

"The profile is then smoothed using a 14-point moving average over the 49-point profile"

--> No justification given anywhere for this. Any other studies doing similar things that you could reference here?

--> You should explain what the purpose of this smoothing is. My impression is that it's not even necessary (see last part of my Major Comment #1 for a better option to compare what's resolved by all datasets).

We thank the reviewer for pointing out these methodological weaknesses. We now use the tropopause height derived directly from the ERA5 dataset for all analyses to ensure consistency. The use of NCEP reanalysis has been removed from the manuscript. The 14-point moving average smoothing step has been removed. We agree it was not well-justified. The new band-pass filtering approach (as suggested in Major Comment #1) is a more appropriate and sufficient method for conditioning the perturbation profiles. The Methods section has been updated to reflect these important changes.

(lines 225-227)

**The lower bound is set one kilometer below the tropopause to focus on events extending beyond it, balancing Aeolus' resolution with our interest in upper-end dynamics. For consistency, the tropopause height is derived directly from the ERA5 dataset for all analyses. The profile is then averaged over the selected range, representing the Ek, as seen in Fig.1c.**

#

**3: treatment of GNSS-RO data, details not properly justified**

#

l.206-211: details are very vague, e.g. which windowing is used for both in the end? Please state clearly what settings are applied to RO data and Aeolus.

l.213-214:

"Where the Brunt-Vaisala frequency squared (N2) is smoothed using binomial (Gaussian) smoothing of 10th order."

--> This is not justified, where is this coming from? Any reference for this?

"Consequently, the data treatment across various instruments, whether wind or temperature remains consistent"

--> I strongly disagree!

We apologize for the lack of clarity and justification in this section. We have clarified in the manuscript that the same Welch windowing function was applied to all perturbation profiles from all datasets. We have removed the 10th-order binomial smoothing of the $N^2$ profile. We thank the reviewer for this critical comment and agree that this step was not standard practice. As argued in studies like Alexander et al. (2008b), the spatio-temporal averaging used to derive the background state provides a sufficiently smooth and stable background temperature profile for the $N^2$ calculation. With these changes, our statement about the consistency of data treatment is now properly supported. The section has been revised for clarity and accuracy.

(lines 237-240)

**The main difference lies in substituting temperature T(z) for wind U(z) throughout the background-perturbation decomposition [...] The Welch window was applied to all perturbation profiles (wind and temperature) before filtering to mitigate spectral leakage. The same band-pass filtering strategy and vertical averaging then provide the Ep profile from the temperature perturbations.**

#

**4: Most figures are low quality**

#

Fig.2:

--> please use the degree sign ° and not "DegN/E" (also present in Figs.4,5,6,7... all figures with lon or lat dimension...)

--> color scale is not the best for visibility, a color every 0.1 would improve guessing the exact value by eye.

--> It appears a bit pixelated if one zooms in just a bit.

Fig.3

--> unreasonably large to show only four lines

--> way too many labels on the x-axis

Fig.5

--> label sizes too small

--> too many labels on y-axis

--> odd alignment of a)b)c) with subpanel titles

--> b) panel title size mismatch with the others, looks like put there by hand unlike a) and c)

The figures have been reworked to be as high quality as possible, with the according fixes applied to fig 2 and 3 (now in the appendix) as well as the geographical maps and hovmoller plots. We also ensure that degree signs, label sizes, and colorbar intervals follow the specific corrections the reviewer has suggested.

We acknowledge that the apparent pixelation or blurriness described by the reviewer. Such issues arise primarily from the PDF conversion process currently used to assemble the manuscript draft. Despite multiple attempts, it has been difficult to completely avoid compression artifacts when exporting to PDF from Word, especially for complex maps and plots with fine detail. We are deeply sorry for the inconvenience this has caused in the review process. We would like to reassure the reviewer that these issues will not occur in the final submission: for the production-ready version we will provide all figures separately as individual high-resolution vector or high-quality raster files (following Copernicus/EGU guidelines on figure preparation).

#

**5: pages 11-14**

#

-whole pages 11-12: this can be briefly summarized in the main manuscript and all the details moved to a supplement, including Fig.3

-The noise correction makes a quite long list of assumptions, could the authors provide some results/comparison of Aeolus results without noise correction for reference?

-Fig.4: the stark contrast of MAM 2019 and MAM 2020 does not give a reader a lot of confidence in your method. I am skeptical of how realistic the evolution of the left column is (the noise-corrected AEOLUS HLOS*).

-l.351-352: "The geographical distribution and evolution of energy hotspots are largely similar between the two datasets"

--> I disagree, the evolution of their strength, even in relative terms, seems quite different: e.g. compare the last 3-4 rows.

We acknowledge the concerns about the original results. A primary reason for the "stark contrast" between years was a flaw in our original noise correction. We have developed a much more robust, spatio-temporally adaptive algorithm (detailed in Appendix D). All results and figures have been regenerated with this new method, along with the new vertical grid and filtering. The new results no longer exhibit the unrealistic jumps between seasons. We have moved the full mathematical derivation and validation plots (including the original Figure 3) to a new, comprehensive Appendix D. The main text now contains a concise summary of the method's principles. Appendix D also now includes a plot of the uncorrected Aeolus data for reference, as requested. With the new results, we have completely re-written the interpretation in Section 3.1.

#

#

**Minor / technical comments**

#

#

**Abstract**

"revealing opportunities to refine reanalysis products and model parameterizations, as well as improving the energy ratio."

--> too vague, is there any specific recommendation here?

The following paragraph has been modified

**The combination of Aeolus and GNSS-RO data allows for an observationally-based examination of the partitioning between kinetic and potential energy, highlighting discrepancies with reanalysis products that could inform future model parameterization development.**

-l.2: cite ERA5 reference here

Done

-l.28-30: but the Podglajen study is from before ERA5 was around, please rephrase sentence for consistency.

Done

-l.45-46: "short-wavelength waves are primarily lower frequency gravity waves, as dictated by the dispersion relation"

--> To avoid confusion please specify that it's short vertical wavelength, and give a ballpark number of the range of vert. wavelengths you are referring to.

--> Also in the next sentence, specify what vertical wavelenghts can be captured by Aeolus.

The following paragraph has been updated

**These waves with short vertical wavelengths (typically 2-10 km) are primarily lower-frequency gravity waves, as dictated by the dispersion relation, and exhibit relatively large amplitude wind variability. The Aeolus satellite, equipped with its Atmospheric LAser Doppler INstrument (ALADIN), is able to measure global wind profiles up to an altitude of 30 km, providing insights into the behavior of gravity waves with vertical wavelengths down to ~1.5-2 km in these critical atmospheric layers (Banyard et al., 2021; Rennie et al., 2021; Ratynski et al., 2023).**

-l.59-61: calling it "climatology" from 3 years sounds a bit stretched...

--> perhaps simply state this as an observational estimate for Jun.2019-Aug.2022

--> also this is an example of a repetitive sentence. E.g. 'and its link with deep convection' could be removed without any loss of information

The sentence has been changed to :

**By comparing direct measurements with ERA5 data, we reveal certain limitations in the reanalysis's ability to represent tropical gravity wave dynamics. We will look at the most recent reprocessed Aeolus baseline 2B16, providing data from June 2019 to August 2022**

And the other sentence was removed

-l.70: I would support sub-subsections for each separate dataset and methods.

We acknowledge the suggestion to introduce sub-sections for each dataset to improve readability. However, since all datasets are now mapped to ERA5's resolution, and the GNSSRO treatment has been streamlined to match Aeolus, there are effectively only two types of processing. We believe that organizing the methods by Ek/Ep categorisation is preferable, as it avoids unnecessary repetition between ERA5 and its counterparts and keeps the text more concise.

-l.72: range bin settings and other specifications should have an earlier reference.

--> Also please update the Rennie and Isaksen 2020 reference to the 2024 ESA contract report (which includes all information from the 2020 TM). Check throughout the manuscript.

--> https://www.ecmwf.int/en/elibrary/81546-nwp-impact-aeolus-level-2b-winds-ecmwf

The reference for these details is now prominently placed at the end of this introductory block, ensuring that all preceding technical information is immediately and clearly sourced. We have also updated the reference to the latest 2024 ECMWF report by Rennie and Isaksen throughout the manuscript, as requested.

-l.87-88: please confirm whether you got the data on that native resolution?

The reviewer is correct to point out the distinction between the model's native grid and the data product we use. The data products we downloaded from the ECMWF archive were pre-interpolated onto a regular 0.25° x 0.25° latitude-longitude grid. We have revised the manuscript to state this explicitly and avoid any ambiguity.

-l.94: best candidate (by far in my opinion), especially when compared to other reanalysis products.

The sentence has been reworded as "the best candidate".

-l.95: "standard" --> you mean the 137 hybrid levels? Standard is usually associated with the 37 standard pressure levels, I recommend not using this term here to avoid confusion.

We confirm that ERA5 data was retrieved on native 137 model levels, not "standard" levels. Text corrected

-l.99: GNSS-RO datasets --> please list which missions are included + their references, and I presume COSMIC-2 dominates the overall data amount? If so, mentioning a bit

COSMIC-2/FORMOSAT-7 and other third-party RO missions were evaluated and monitored by ROM SAF (several technical reports exist), but no routinely generated ROM SAF products based on those data were disseminated in the 2019-2022 operations reports. The satellites used in this analysis are the Metop constellation (Metop-B & Metop-C continuously and Metop-A up to 15 Nov 2021). Refences have been added.

(lines 108 -114)

**For the study period of June 2019 to August 2022 these datasets are dominated by the Metop constellation: Metop-B and Metop-C throughout, with Metop-A contributing until its retirement in November 2021 (von Engeln et al., 2011). These datasets are derived from the bending angles of GNSS signals as they pass through the Earth's atmosphere and are observed by low Earth-orbiting satellites. It provides global coverage with a high vertical resolution, sub-Kelvin accuracy, full diurnal coverage, and all-weather capability. The vertical resolution of GNSS-RO temperature profiles is fundamentally limited by diffraction and varies with altitude, typically ranging from ~0.5 km in the lower troposphere to ~1.4 km in the middle atmosphere (Kursinski et al., 1997)**

-l.111-113: perhaps merge with l.97-98 at the beginning of the paragraph, otherwise to me feels a bit repetitive.

Done

-l.103-104: defined by bending angle gradient, which increases near inversion layers / humidity gradients

--> I recommend to refer to Kursinski et al. 1997 here --> https://doi.org/10.1029/97JD01569

The reference has been added.

-l.115-118: feels very repetitive and could be streamlined

Removed as it added not information

-l.130-131: overselling and too vague, remove or specify recommendations to enhance reanalyses and models from the results of your study.

The sentence has been removed

-l.131-133: just say they are independent datasets, this sentence can be streamlined and toned down.

The sentence has been simplified.

############### 2.2 Methods and limitations

-l.139-150: regarding the trickiness of background state removal, I miss a discussion about research that used GNSS-RO and Aeolus to study Kelvin waves, their vertical scales and (in the case of Randel et al., 2021) the behavior of the small-scale residual.

These references are very relevant to your study's methodology, the more so since they use the same datasets as you.

--> Randel and Wu (2005) --> https://doi.org/10.1029/2004JD005006 (using GPS-RO)

"Vertical wavelengths of ~6–8 km are observed near and above the tropopause in December 2001 to January 2002 (Figures 6a and 6b), while shorter vertical wavelengths (~4–5 km) are observed in May and August–September 2002 (Figures 6c and 6d). "

--> Randel et al. (2021) --> https://doi.org/10.1029/2020JD033969 (using COSMIC-2)

"strong residual variance occurs in the longitudinal shear zones of Kelvin waves" and this small-scale residual T variance is associated with GWs.

--> Zagar et al. (2021) --> https://doi.org/10.1029/2021GL094716 --> "Aeolus assimilation modifies the representation of vertically propagating Kelvin waves in the tropical UTLS" (Aeolus)

A discussion encompassing previous research, the limits of current methods and the excepted caveats has been implemented.

(lines 136 – 166)

The following section discusses the retrieval of GW kinetic energy, Ek. A primary challenge in this retrieval, particularly in the tropical UTLS, is the robust separation of GWs from other dominant, synoptic-to-planetary scale equatorial waves, such as Kelvin waves. Observational studies using GNSS-RO data have consistently shown that Kelvin waves, with typical vertical wavelengths in the range of ~4-8 km (Randel et al., 2021; Randel and Wu, 2005), are a prominent feature of the tropical temperature and wind fields. This presents a potential for spectral overlap with the longer vertical wavelength portion of the GW spectrum that this study aims to capture.

**[…]**

The separation of the wind or temperature profile into a background state and perturbations using HD is intended to isolate fluctuations characteristic of gravity waves by filtering out larger-scale and slower-evolving processes like the mean components of Rossby and Kelvin waves. This selection relies on the distinct scale and structural characteristics of GW perturbations. However, the work by Randel et al., (2021) using dense COSMIC-2 RO data reveals further complexities. They found that "residual" small-scale temperature variances (analogous to our perturbation fields) exhibit coherent maxima in the longitudinal and vertical shear zones of large-scale Kelvin waves. This suggests that the local atmospheric environment shaped by Kelvin waves, particularly variations in static stability ($N^2$), can modulate the amplitude of smaller-scale variability, potentially including GWs. Furthermore, the assimilation of Aeolus wind data itself has been shown to directly impact the representation of vertically propagating Kelvin waves in numerical weather prediction models, especially in regions of strong vertical wind shear (Žagar et al., 2021). This implies that Kelvin waves are indeed present in the Aeolus observations and that their characteristics might differ from those in reanalyses not assimilating Aeolus data.

################

-l.160-161: "Aeolus now provides the necessary tools to apply the same approach for GW Ek."

--> Sorry to be picky here, but what tools does Aeolus bring now that it didn't before. You use the same approach (your tool) to calculate GW Ek from Aeolus (data, not a tool). Such phrasing is just unnecessary verbose.

We reworded the sentence to remove unnecessary verbose speech.

- Fig.1: please include the U(z) notation in the labels, and specify which datasets you take U from.

The figure has been updated.

-l.216-219: you should state all this when introducing Ek{hlos}.

--> Fig.2 belongs in a supplement

The text has been moved up next to the introduction of Ek_hlos and the ex-Fig.2 has been moved to the Appendix.

--> And wouldn't it be fairer to compare EK from ERA5 U to EK_HLOS??

The proposed comparison would involve examining ERA5 zonal kinetic energy (Ek_u_ERA5) against ERA5 HLOS-projected kinetic energy (Ek_HLOS_ERA5). That ratio would assess how well the HLOS projection specifically captures the zonal wind component within the model.

However, our analysis in Appendix B (e.g., Fig. 2) had a different objective. We aimed to quantify how representative a quasi-zonal HLOS measurement (like Aeolus's) is for the total gravity wave kinetic energy, which includes both zonal and meridional components. To do this, we calculated the ratio of ERA5's HLOS-projected kinetic energy (Ek_HLOS_ERA5) to its total kinetic energy (Ek_TOTAL_ERA5, including both u′ and v′ components). This approach was intended to estimate the fraction of total kinetic energy captured by a quasi-zonal line-of-sight measurement and to characterize the spatial and temporal variability of gravity wave anisotropy in the ERA5 model. The ratio explicitly highlights regions where meridional wind perturbations are more significant, therefore pointing where an HLOS-only measurement would miss a larger share of the total energy. We believe this provides a necessary context for interpreting the absolute values measured by Aeolus.

-l.228-231: belongs also in a supplement in my opinion

Moved to appendix.

-l.245-249: a lot of verbose here, show the figure in a supplement and move the text there

Moved to appendix.

-l.343-344: another example of verbose.

The sentence has been removed.

---

## Author Response (AR2)

**Reponse to reviewer #2**

We would like to extend our sincere thanks to Reviewer #2 for their continued feedback on this new iteration. We appreciate the commendation that the results feel more robust, and we agree that addressing the remaining points has further improved the paper's clarity and focus.

We have addressed all of the reviewer's comments, which are detailed below.

I commend the authors for addressing all my comments -- which was quite some work! I am satisfied with most of the revisions and I feel the results are very robust now.

I have a number of remaining minor/technical comments, and only a couple of bigger 'complains' from my side. While due to the overall amount of comments it almost sums to another major revision, it all should be nevertheless fast and straightforward to address.

#

**My two remaining complains:**

- 1) The continuous color scale in most figures, would be much easier to interpret and guess the numbers by eye, if the color scale had discrete steps.
- 2) Too much focus and speculation about the MJO and related topics in the discussion, while I can't notice any MJO-like signal in any of the figures.

We agree completely both suggestions. Continuous color scales can indeed be difficult to interpret precisely. We have updated all relevant figures (Figures 2, 3, 4, 5, 6, and 7) to use discrete color scales.

Upon review, we agree that our discussion of the MJO was overly speculative and not directly supported by the temporal resolution of our analysis. The dominant signal in our figures is indeed the seasonal cycle. Accordingly, we have completely rewritten the relevant paragraphs in the Discussion section (previously lines 560-573 and 612-620). All speculative mentions of the MJO have been removed. The revised discussion now focuses squarely on the robust link between the observed gravity wave activity and the seasonal cycle of major tropical monsoon systems. Similarly, we have refocused the discussion on the QBO to be less speculative and more directly tied to recent findings from the Aeolus mission itself, incorporating the suggested literature.

There are numerous specific comments related to 1) and 2) listed below:

#

**Specific comments:**

- I.28: specify the study period.

**Done**

--> same for the 1st paragraph of section 2.1, the analysis time period should be specified also there.

**Done**

- I.29: remove "referred to as SO3 in"

Done

- About the GWD scheme used in ERA5, this should be mentioned again in the 2.1 subsection and in your discussion regarding Fig.3.

We added the following line in the 2.1 subsection:

(lines 106-108)

For representing sub-grid scale gravity waves, the ERA5 configuration used in this study employs a non-orographic GWD parameterization that is not directly forced by model-diagnosed convection (Orr et al., 2010). Instead, the scheme launches a globally uniform spectrum of waves from the troposphere.

And the following line in the discussion:

(lines 425-428)

The difference between the two datasets, shown in Fig.3c, quantifies this discrepancy. The plot is overwhelmingly positive, indicating a systematic and significant underestimation of GW kinetic energy by ERA5 throughout the tropics. The regions of greatest underestimation, where the difference exceeds 10 J/kg, align almost perfectly with areas of deep convection, as identified by the low OLR contours. This last element reinforces the conclusion that ERA5's key limitation lies in its representation of convection-driven wave activity. This finding is consistent with the fact that ERA5's non-orographic GWD scheme is not directly coupled to model-diagnosed convection, highlighting the need for improved parameterizations to better capture these sources.

- I.83-84: perhaps describe more specifics of RBS in your region of study in the previous sentence (like what resolution exactly it has in the UTLS height range), and here just state that RBS are different in the extratropics/polar regions.

We have revised the text to provide the typical vertical resolution for our study region upfront and have clarified that the Range Bin Settings are geographically dependent. The revised sentence now reads:

(lines 83-86)

within the tropical UTLS region of this study, the vertical bin size is typically between 0.5 and 1.5 km. The distribution of these range bins is determined by a dedicated range bin setting (RBS), which varies geographically to meet different observational goals, with distinct configurations routinely used for the tropics, extratropics, and polar regions.

- I.104: interpolated how? (refer to methods subsection)

Thank you for pointing out that the description of our interpolation method was not sufficiently detailed. The revised text now reads:

(lines 108-111)

For this study, wind components are retrieved on the native 137 model levels. To prepare the data for analysis, the geopotential height of each model level is first converted to geometric altitude. The vertical profiles are then linearly interpolated from this native geometric altitude grid onto the standard 100 m high-resolution grid used for all datasets in this study.

Regarding the suggestion to refer to the Methods subsection (2.2), we ultimately chose to include this concise, two-sentence explanation directly within the description of the ERA5 dataset in Section 2.1. We believe this approach provides the necessary clarity for the reader at the moment the dataset is introduced, without requiring them to cross-reference another section for this specific detail.

- I.124-125: reference needed

Added reference to Schmidt et al., 2016 that directly references this method.

- l.132-133: to make the sentence more concise, substitute "is intentional. This... by treating it as" --> serves as a comparison with

**Done**

- l.167-168: Ern et al (2023) proved this, also Zagar et al. (2025) detail more about Kelvin waves and shear.

Reworked the following paragraph:

**(Lines 164-171)**

Furthermore, data assimilation studies have demonstrated that the inclusion of Aeolus wind data directly impacts the representation of vertically propagating Kelvin waves in numerical weather prediction models. This impact is explicitly linked to the background wind, with the largest analysis changes occurring in regions of strong vertical wind shear (Žagar et al., 2021, 2025). This highlights the importance of direct wind observations in these critical regions. Indeed, direct analysis of Aeolus observations (without assimilation) confirms that Kelvin waves are well-resolved, showing good agreement in wave variances when compared to reanalyses (Ern et al., 2023). This implies that the characteristics of Kelvin waves seen by Aeolus are robust and may differ from those in reanalyses not assimilating Aeolus data.

- 1.226-227: "The lower bound is set..." --> sentence can be removed without loss of any info

**Done**

- I.228: tropopause height, specify how it is defined, e.g. cold-point from model levels?

We have removed the redundant sentence and have added a clear definition of how the tropopause height was determined (WMO thermal definition) at lines 235-238.

- I.278-279: cite Lux et al (2022) for this

**Done**

- Figure 2: please improve the color scale, make separate colors every 1 or 2 J/Kg -- i.e. make it discrete, not continuous -- add another color beyond yellow.

**Done**

- I.314: "Boreal spring 2019 to Austral summer 2020" is an extremely awkward way to state your analysis period. Simply JJA 2019 to MAM 2021 does the job, without confusing the reader.

**Done**

- I.315: patterns, describe which ones.

We have explicitly described the two main large-scale patterns observed:

(lines 329-332)

This comparison reveals both key similarities in two large-scale patterns: first, the confinement of most GW kinetic energy to the equatorial belt (approximately 15°S–15°N), and second, a distinct seasonal migration of this energy. However, there are also significant differences in the representation of regional wave activity.

- I.318-319: --> The reader should note that also some variance from equatorial waves, centered at the equator (by definition), will be inevitably present to some small degree.

Added as is at lines 335-337

- I.324-325: In ERA5 a lot is missing especially in the active monsoon regions // further away from the Equator, please note this in the text.

The following text has been added on the following paragraph discussing differences:

(lines 360-362)

ERA5 tends to represent GW activity as a relatively smooth, zonally elongated band, with modest seasonal modulation and appears to significantly miss wave activity both in the active monsoon regions and in more structured events further from the equator.

- I.334-338: Again, I feel seasonal monsoon convection is missing in this discussion.

We agree that the paragraph discussing subtropical jet sources needed to be more clearly contrasted with the convectively-generated gravity waves from tropical monsoon systems. To address this, we have added a concluding sentence to the paragraph that explicitly makes this distinction and clarifies the relative importance of the two sources as indicated by our results. The new sentences read:

(lines 356-359)

These jet- and front-generated waves are dynamically distinct from the deep tropical convection associated with the major seasonal monsoon systems. While the subtropical jets produce notable GW activity, our results indicate that the most intense and geographically extensive hotspots are found within the equatorial belt and are closely tied to these monsoon systems (Kang et al., 2017; Wright and Gille, 2011).

- I.349-356: Perhaps mention that this will be looked in more detail in the next subsection in relation to OLR.
- --> Figure 3 confirms your GW hotspots follow deep convective systems

We have added a sentence at lines 378-380 to transition the reader to the next section

- Figure 3:
- --> Especially in panels a-b, I see the same problem as with Fig.2: please make the color scale discrete (in every panels would be best)

**Done**

--> Longrange --> Longwave!

**Done**

--> I think there might be some mask where positive values (red) are not completely transparent to the stippling, so it appears light gray.

Stippling within blue regions looks fine.

I assume the stippling is intended to be the same everywhere

**Fixed**

- I.373: "the observations" --> more precision needed here -- Aeolus and ERA5 (HLOS) GW and their difference.

**Done**

- I.375-377: need to mention the OLR patterns that coincide with your GW hotspots

The new text now reads:

(lines 403-404)

This migration of high Ek is systematically co-located with the seasonal cycle of convection, with the hotspots consistently falling within the low OLR contours (below 220 W/m²).

- l.386-388: again your figure shows this is the case -- GW hotspots following low OLR regions, please discuss this and state the OLR values shown in the figure.

The revised paragraph now includes the following sentence:

(Lines 412-414)

The strong spatial correlation shown in Figure 3a between the most intense kinetic energy observed by Aeolus and the lowest OLR values (< 210 W/m²) provides evidence that these mechanisms are the primary drivers of the observed GWs.

- I.397-399: Better to simply state it's a good estimator of cloud top temperature and thereby convection depth.

We have adopted the more concise physical description of OLR as suggested. Furthermore, we recognized that explaining OLR after having already discussed its patterns was structurally awkward. Therefore, we have moved this new, concise definition to the beginning of Section 3.2, where the OLR contours are first introduced to the reader. The text now introduces OLR as:

- a reliable proxy for deep convection as it indicates cold, high-altitude cloud tops and thus the depth of convective systems (Zhang et al., 2017).
- I.400: "primary weakness"
- --> not sure about calling it a primary weakness...

These convective GW's are not parameterized basically, from what I understand from your introduction?

This result from Fig. 3 highlights the need to parameterize convectively generated GWs better.

The former paragraph got reworked. We agree that using the word weakness was misleading as ERA5 does not have a proper parameterization:

(lines 425-428)

The difference between the two datasets, shown in Fig.3c, quantifies this discrepancy. The plot is overwhelmingly positive, indicating a systematic and significant underestimation of GW kinetic energy by ERA5 throughout the tropics. The regions of greatest underestimation, where the difference exceeds 10 J/kg, align almost perfectly with areas of deep convection, as identified by the low OLR contours. This last element reinforces the conclusion that ERA5's key limitation lies in its representation of convection-driven wave activity. This finding is consistent with the fact that ERA5's non-orographic GWD scheme is not directly coupled to model-diagnosed convection, highlighting the need for improved parameterizations to better capture these sources.

- I.410-411: Either be more specific about what metrics you refer to, or remove this sentence (no info lost if removed)

**Sentence removed**

- Figure 4: Please improve color scale with a discrete separation of colors every 0.1.

As it is now, it's impossible to tell by eye where exactly the 2 ratio is, or whether it's 1.8 instead.

As shown in these two examples, we decided that a 0.25 discrete separation served the figure better than 0.1, which was still quite hard to distinguish.

- I.423-424: the caption about OLR --> Can be shortened to "white and black contour lines represent 210 and 220 W/m2 OLR, respectively."

Updated all captions except for the first one on figure 3.

- I.438: Oscillations --> Oscillation (no plurals with MJO)

**Done.**

- I.447: dynamically active, you mean convectively?

Yes, we added that precision.

- I.456: each line--> each row

**Done.**

- I.457: I don't see any white bins

Removed that sentence.

- Figures 5-6-7: same as previous figures, please make a discrete color scale.

Done.

- I.489: "lesser convective areas" --> you mean non-convective?

Perhaps consider adding a contour line with high OLR indicating stable conditions into Fig. 6.

Changed the wording and added the high OLR contour.

- I.506-512: I'd like to see a longer discussion comparing it with Fig. 3 in terms of general values, peaks and sign of the differences
- --> Good agreement with ERA5 in Ep, but I see lots of (light red) color in Fig.6c.
- --> Poor agreement with ERA5 in Fig.3 --> clear underestimation of Ek in convective regions (red colors), but what about the general blue color (although insignificant?) elsewhere.

The following paragraph replaced the older one, proposing a more detailed discussion on the comparison between both figures, addressing the quantitative discrepancies:

(lines 527-541)

The differences between ERA5 and GNSS-RO data, depicted in Fig. 6c, show a mean absolute difference of 1.96 J/kg. This reflects a reasonable agreement, given that ERA5 assimilates GNSS-RO measurements. While there is a slight positive mean bias of 1.68 J/kg (GNSS-RO > ERA5), which accounts for the prevalence of light red colors in the plot, the differences are scattered and show no large-scale, systematic pattern correlated with convection. This stands in stark contrast to the systematic and large discrepancies observed in the kinetic energy fields.

The Ek differences are not only larger in magnitude, with a standard deviation nearly twice that of Ep (3.16 J/kg vs. 1.82 J/kg) and a maximum underestimation by ERA5 that is almost three times greater (>24 J/kg vs. ~9 J/kg for Ep), but they are also structurally different. The Ek difference plot is dominated by large, cohesive regions of statistically significant positive values (red), indicating a systematic underestimation by ERA5. While some areas do show a negative difference (blue color), these are of small magnitude and, as confirmed by the lack of stippling, are not statistically significant. Most importantly, the peak underestimation of Ek is systematically co-located with the deepest convective regions (inside the low OLR contours), whereas the minor differences in Ep show no such alignment. Taken together, this evidence points to a specific limitation in the reanalysis: the issue is not a general failure to represent wave energy, but a targeted inability of the model's physics and data assimilation system to generate the intense, localized kinetic component of gravity waves originating from strong convection in data-sparse regions.

-I. 535: "including orographic influences": there is no orography at 200 lon

Changed to "the distribution of large land masses"

- I.541: "RO temperature data" --> maybe simply state "RO measurements".

what is actually being assimilated are GNSS-RO bending angles (which contains the temperature information within).

Changed to "GNSS-RO measurements, specifically bending angles which contain temperature information"

- 1.558: to me inconsistent implies lots of ups and downs.
- --> perhaps decaying performance is more suitable here?

Yes, changed.

- I.564-565: Seasonal cycle or monsoon could be as well.
- I.560-573: Whole MJO discussion is very speculative and not backed by any result of yours. What one can see clearly in Figs. 2 and 3 is the seasonal cycle.

MJO has a timescale of 30-90 days, meaning full positive-negative phase within 3 months at most, moving form the Indian Ocean into the Pacific: sorry but I don't see anything on that timescale in these figures.

We agree that our original discussion overemphasized the role of the MJO, which is not clearly resolved in our analysis, and that the dominant signal in our figures is indeed the seasonal cycle. Following this advice, we have completely removed the speculative discussion about the MJO. We have rewritten the entire paragraph (previously lines 560-573) to focus squarely on the link between the observed gravity wave activity and the seasonal cycle of the major tropical monsoon systems, which is strongly supported by our figures and a comprehensive body of literature. The revised paragraph now reads:

(lines 589-602)

The analysis of ALADIN wind profiling and ECMWF ERA5 reanalysis data, provided in Fig.2 and Fig.3, revealed enhanced GW activity over the Indian Ocean during Boreal Summer, as well as over the western Pacific and maritime continent in Boreal Winter. The migration of this enhanced GW activity from eastern Africa to the Pacific maritime continent follows a clear seasonal cycle, strongly linked to deep convection as shown by the correlation with regional OLR minima. This robust seasonal pattern indicates that the underlying wave sources are organized by planetary-scale phenomena, primarily the major tropical monsoon systems (Wright and Gille, 2011). The structures observed by Aeolus are therefore highly consistent with the kinetic energy signature of gravity waves generated by the powerful thermal and mechanical forcing mechanisms (Beres et al., 2005; Corcos et al., 2025) known to occur within the large, organized convective systems of the Asian, African, and Maritime Continent monsoons (Kang et al., 2017; Liu et al., 2022). Previous satellite climatologies have firmly established these monsoon regions as dominant global hotspots for stratospheric gravity wave activity (Hindley et al., 2020; Wright and Gille, 2011). This suggests that Aeolus is effectively capturing these seasonally-driven, convection-induced GWs that are underrepresented in ERA5. One of the persistent features observed throughout the study was the high-energy gravity wave hotspot over the African continent, which remained consistent across seasons and years. This suggests a continuous mechanism of continental convection driving gravity wave activity in this region.

- I.574-583: this is actually an important result and should belong in an earlier result section and not here.

Expand the discussion in lines 230-235 with the infos from this paragraph.

You may refer to this in the discussion/conclusions in a summarized manner later.

The following text was added to directly explain the results:

(Lines 244-249)

By shifting the analysis layer upward to such levels, we confirm that the geographical patterns of the energy hotspots are remarkably stable (spatial correlation r > 0.83), and that the vast majority of the peak energy (~88-91%) persists well into the stratosphere. If the signal were dominated by shallow tropospheric outflow, the energy peaks would have collapsed when the analysis layer was moved above the tropopause. The fact that a strong, structured signal remains confirms that our method is observing vertically propagating gravity waves that have penetrated the lower stratosphere.

And the discussion section element was shortened and integrated at the start of the next paragraph: Having established that the Aeolus kinetic energy signal is robust and represents vertically propagating stratospheric gravity waves rather than tropospheric artifacts (as confirmed by our sensitivity analysis in Sect. 2.2), we can use external information to arbitrate the cause of the discrepancy with ERA5.

- l.601-607: Newer papers from Zagar on Aeolus and equatorial waves (2021, and especially 2025) would help with this discussion.

The IFS has evolved quite a lot from 2004.

We agree that our discussion needed to be updated to reflect the evolution of the IFS and to incorporate the latest findings from the Aeolus mission. To address this, we have revised the paragraph by adding the newer references to demonstrate that this issue remains relevant today: (lines 622-632)

These statistical relationships are primarily designed to represent large-scale, quasi-balanced (rotational) flow and have long been known to be less effective at specifying the smaller-scale, divergent component of the wind field to which convectively generated gravity waves belong, especially in the tropics (Žagar et al., 2004). While the Integrated Forecasting System (IFS) has evolved considerably, recent Observing System Experiments (OSEs) using Aeolus data confirm that this challenge persists. These studies provide direct evidence that the assimilation of Aeolus wind profiles systematically enhances the analyzed amplitudes of equatorial waves, particularly in regions of strong vertical wind shear where the model's background state is most uncertain (Žagar et al., 2021, 2025). Consequently, while the assimilation of GNSS-RO constrains the thermodynamic (Ep) aspect of the wave, the system lacks the necessary information and dynamic constraints to generate the corresponding divergent wind perturbations, leading to the observed Ek deficit. This process evidently fails to capture the full spectrum of high-Ek wave modes generated by convection.

- I.612-620: too speculative, was there any remarkable MJO event during your analysis time period? I again suggest to focus the discussion more on seasonal monsoon convection, and existing results on Aeolus capturing equatorial waves (e.g. the previously mentione Ern and Zagar papers), and the QBO (Banyard).

Also, spending almost an entire paragraph on QBO-MJO modulation is going off-topic.

We thank the reviewer for this very relevant feedback. We agree that our original paragraph on QBO-MJO modulation was too speculative and distracted from the main arguments of the paper. Following this advice, we have completely removed the original paragraph. In its place, we have written a new, more focused paragraph that discusses the known limitations of ERA5 in the context of the QBO, directly incorporating the highly relevant recent literature you suggested:

(lines 637-649)

These challenges are particularly evident in the representation of key tropical phenomena like the Quasi-Biennial Oscillation (QBO), which is driven by the upward propagation and dissipation of a spectrum of atmospheric waves. Recent studies using direct Aeolus wind observations have provided new insights into how reanalyses represent these processes. For instance, Banyard et al. (2023) found that during the 2019/2020 QBO disruption, a period covered by our study, the onset of the disruptive easterly jet was observed by Aeolus five days earlier than in ERA5. This discrepancy was linked to higher Kelvin wave variances and sharper vertical wind shear in the Aeolus data, suggesting that ERA5 may misrepresent the breaking of smaller-scale waves that are crucial for forcing the QBO. Similarly, Ern et al. (2023) confirmed that while the zonal-mean QBO is well-represented in ERA5, local biases exist, particularly in shear zones. From a data assimilation perspective, Žagar et al. (2025) showed that assimilating Aeolus winds produced the largest changes to the analyzed state in the UTLS precisely during the 2019/2020 QBO disruption, highlighting the importance of direct wind observations for reducing uncertainties in these critical shear zones. Together, these findings, derived from the same novel wind dataset used here, support our conclusion that reanalyses can have significant deficiencies in representing the full spectrum of wave activity and its associated kinetic energy in the absence of direct wind assimilation.

- l.632-633: No works that are more recent? It's surprising to me, that the newest reference here is 10y old and that there were no follow-ups (although I'm not an expert in this particular field)

Works from 2021, 2022 and a very recent 2025 paper were added to actualize both sides of the argument with newer studies:

(lines 654-671)

At first glance, using a fixed ratio appears straightforward for converting well-documented Ep (from temperature-based instruments such as GNSS-RO) to Ek. Traditionally, linear GW theory proposes a near-constant ratio of Ek to Ep, often quoted between 5/3 and 2.0 (VanZandt, 1985; Hei et al., 2008). In idealized models of linear wave behavior, the kinetic and potential energies are expected to be comparable, leading to a ratio close to unity. This theoretical relationship has been confirmed observationally. In stable, linear wave

conditions, the energy ratios adhere closely to predictions (Nastrom et al., 2000), a finding supported by a modern case study of individual, freely-propagating waves (Huang et al., 2021).

However, a growing body of evidence challenges this simplification: Empirical work increasingly reveals significant variability in this ratio, indicating non-linear effects in real-world atmospheric conditions (Wing et al., 2025; Baumgarten et al., 2015; Guharay et al., 2010; Tsuda et al., 2004). When the observed energy ratios deviate significantly from this expected range, non-linear processes may be at play. While a large climatological study may find a mean Ek/Ep ratio close to theoretical values (e.g., 1.5 in Zhang et al., 2022), this average can mask significant event-to-event variability. For instance, in situations where wave amplitudes are particularly large, wave-wave interactions, such as those resulting from wave breaking or saturation, could lead to the observed discrepancies. This has been demonstrated in earlier work by Mack and Jay. (1967), who found that under certain conditions, potential energy deviated markedly from kinetic energy, suggesting non-linear effects. Similar findings have been reported by Fritts et al. (2009), who showed that interactions between gravity waves and fine atmospheric structures can result in turbulence, thereby affecting the balance between kinetic and potential energy. A recent study also confirmed that the ratio is not static and can be actively modulated by the background atmospheric state, such as strong wind shear (Wing et al., 2025).

- I.645: "increasingly challenged by observations" --> references please, or say "(see references above / previous paragraph)"

Added "(see references in the previous paragraph)".

- l.652-653: refer to the section where you come up with these numbers

Added "(as detailed in Sect. 2.2 and shown in Appendix C)".

- I.657-665: regarding the hot pixels, wasnt this taken care of in the latest reprocessing that you use?
- Even if you state later that the latest baseline improves this, the current phrasing implies it's still a major source of uncertainties -- which is not really the case right?
- --> I overall don't like the fit of this sentence for the study's research focus, I suggest to remove even the whole paragraph -- it's mixing many things in a very vague way, without any way forward.
- --> Oscillating perturbations misiterpreted as GW signals (Ratysnki et al 2023) can be mentioned in the Data/Methods section to give more nuance on the reliability of the dataset.

The paragraph was removed, there is already a sentence talking about OPs in the original data section.

- I.676: include lower bound of the wavelengths that are band-pass filtered. Also, band-pass is the more common expression (instead of passband)

**Done.**

- I.687-693: please also mention Aeolus' swath size vs whole globe -- small scales are only sampled very locally every day --> how many equator crossings every 12h, I believe it's in the range of 14-16?

Thank you for the suggestion. We have now added a discussion of Aeolus' narrow observational geometry and its implications for sampling small-scale gravity waves. The revised paragraph explicitly states the ~3 km effective swath width, the ~86 km along-track averaging, and the ~16 orbits per day (~15–16 equator crossings every 12 h). Here's the revised paragraph:

(Lines 710-721)

Looking forward, a critical application for such observations is the constraint of gravity wave momentum fluxes, which are essential for global circulation models. However, deriving momentum flux estimates directly from single-component wind measurements like those from Aeolus presents two co-dependent problems. First, the vertical flux of horizontal momentum (e.g., (u'w')) requires simultaneous knowledge of horizontal (u') and vertical (w') wind perturbations. Aeolus supplies only the line-of-sight projection of the horizontal wind and, crucially, no direct information on the vertical wind. In the standard processing w' is simply assumed negligible (Krisch et al., 2022), leaving the key term in the flux equation unconstrained. Second, the satellite's sampling geometry further limits what can be inferred. Aeolus observes with a ~3 km-wide "pencil beam" that is horizontally averaged to about 86 km along track, and its sun-synchronous orbit completes ~16 revolutions per day (roughly 32 equator crossings, or 15–16 every 12 h). Small-scale gravity waves are therefore captured only where the narrow ground tracks happen to intersect them, leaving large spatial and temporal gaps. Together, the absence of direct w' measurements and this sparse, one-dimensional sampling mean that Aeolus winds alone cannot yield global momentum-flux maps without substantial modelling support or complementary observations.

**# References**

#

Ern et al (2023), The quasi-biennial oscillation (QBO) and global-scale tropical waves in Aeolus wind observations, radiosonde data, and reanalyses, https://doi.org/10.5194/acp-23-9549-2023

Lux et al. (2022), Quality control and error assessment of the Aeolus L2B wind results from the Joint Aeolus Tropical Atlantic Campaign, <a href="https://doi.org/10.5194/amt-15-6467-2022">https://doi.org/10.5194/amt-15-6467-2022</a>

Zagar et al. (2025), ESA's Aeolus mission reveals uncertainties in tropical wind and wave-driven circulations, http://dx.doi.org/10.1029/2025GL114832